# Hyper-parameter Tuning for Fair Classification without Sensitive Attribute Access

**Akshaj Kumar Veldanda**                                                                 *akv275@nyu.edu*
*Electrical and Computer Engineering Department*
*New York University*

**Ivan Brugere**\*                                                                        *ivan.brugere@jpmchase.com*
*JP Morgan Chase AI Research*

**Sanghamitra Dutta**\*                                                                   *sanghamd@umd.edu*
*Electrical and Computer Engineering Department*
*University of Maryland College Park*

**Alan Mishler**\*                                                                        *alan.mishler@jpmchase.com*
*JP Morgan Chase AI Research*

**Siddharth Garg**                                                                        *sg175@nyu.edu*
*Electrical and Computer Engineering Department*
*New York University*

**Reviewed on OpenReview:** *https://openreview.net/forum?id=ZSWKdRi2cU*

## Abstract

Fair machine learning methods seek to train models that balance model performance across demographic subgroups defined over sensitive attributes like race and gender. Although sensitive attributes are typically assumed to be known during training, they may not be available in practice due to privacy and other logistical concerns. Recent work has sought to train fair models without sensitive attributes on training data. However, these methods need extensive hyper-parameter tuning to achieve good results, and hence assume that sensitive attributes are known on validation data. However, this assumption too might not be practical. Here, we propose Antigone, a framework to train fair classifiers without access to sensitive attributes on either training or validation data. Instead, we generate pseudo sensitive attributes on the validation data by training a ERM model and using the classifier's incorrectly (correctly) classified examples as proxies for disadvantaged (advantaged) groups. Since fairness metrics like demographic parity, equal opportunity and subgroup accuracy can be estimated to within a proportionality constant even with noisy sensitive attribute information, we show theoretically and empirically that these proxy labels can be used to maximize fairness under average accuracy constraints. Key to our results is a principled approach to select the hyper-parameters of the ERM model in a completely unsupervised fashion (meaning without access to ground truth sensitive attributes) that minimizes the gap between fairness estimated using noisy versus ground-truth sensitive labels. We demonstrate that Antigone outperforms existing methods on CelebA, Waterbirds, and UCI datasets.

## 1   Introduction

Despite their success on a range of real-world tasks, prior work (Hovy & Søgaard, 2015; Oren et al., 2019; Hashimoto et al., 2018a) has found that state-of-the-art deep neural networks exhibit unintended biases

---

\*Equal Contribution

towards specific subgroups, for instance towards disadvantaged subgroups or because of tendency to learn spurious correlations and simplicity bias, especially harming disadvantaged groups. Seminal work by Buolamwini & Gebru (2018) demonstrated, for instance, that commercial face recognition systems had lower accuracy on darker-skinned women than other groups. A body of work has sought to design fair machine learning algorithms that account for a model's performance on a per-group basis (Prost et al., 2019; Liu et al., 2021; Sohoni et al., 2020).

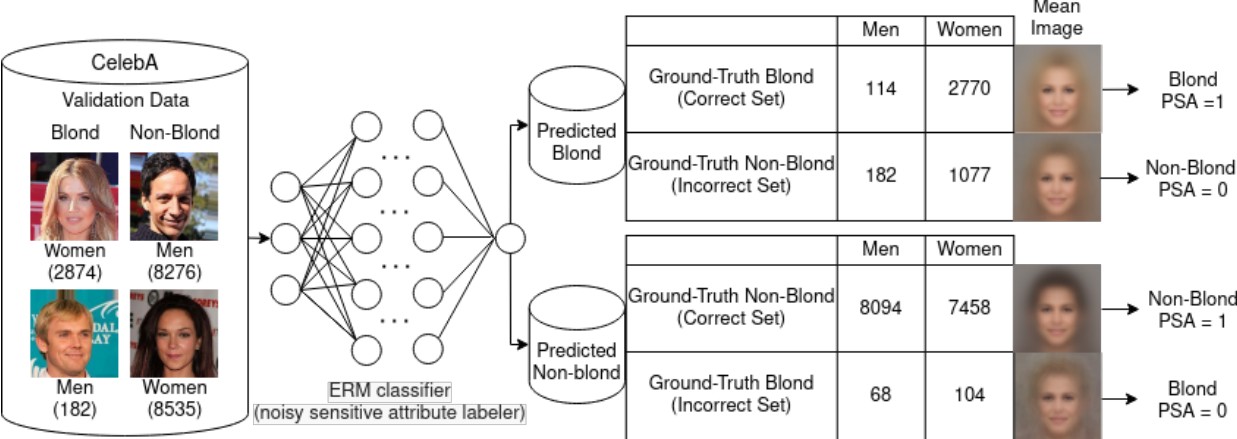

Figure 1: Antigone on CelebA dataset with hair color as target label and gender as (unknown) sensitive attribute. Blond men are discriminated against. Correspondingly, the mean image of the Blond class incorrect set (row 4) has more male features than that of its correct set (row 1), reflecting this bias. Similarly, a bias against non-blond women is also reflected. PSA = 0 corresponds to disadvantaged groups, and PSA = 1 corresponds to advantaged groups.

Prior work typically assumes that attributes, *e.g.* gender and race, on which we seek to train fair models are available on training and validation data (Sagawa* et al., 2020; Prost et al., 2019). We will refer to these as *sensitive attributes* (SA). However, recent work (Veale & Binns, 2017; Holstein et al., 2019) has highlighted many real-world settings in which SA may not be available. For example, data subjects may abstain from providing sensitive information for privacy reasons or to evade future discrimination (Markos et al., 2017). Attributes on which the model discriminates might not be known or available during training and only identified post-deployment (Citron & Pasquale, 2014; Pasquale, 2015; Veldanda et al., 2023b). For instance, recent work shows that fair NLP models trained on western datasets discriminate based on last names when re-contextualized to geo-cultural settings like India (Bhatt et al., 2022). Similarly, reports suggest that Nikon's face detection models repeatedly identify Asian faces as blinking, a bias that was only identified retrospectively (Leslie, 2020). Unfortunately, by this point, at least some harm is already incurred.

Recent work has therefore sought to train fair classifiers *without* SA on the training set (Liu et al., 2021; Creager et al., 2021; Nam et al., 2020; Hashimoto et al., 2018a). At their core, these methods first identify advantaged and disadvantaged groups within the training dataset; i.e., subgroups with high and low accuracy, respectively. Then, they deploy various training strategies to boost performance on disadvantaged groups. However, these methods are *highly* sensitive to the choice of hyperparameters, and can actually *hurt* fairness compared to baseline empirical risk minimization (ERM) without proper hyperparameter tuning (Liu et al., 2021). Hence, with the exception of GEORGE (Sohoni et al., 2020) and ARL (Lahoti et al., 2020), prior work has assumed access to SA on the validation data. However, in practical settings, SA may not be available for the same reasons they are unavailable on training data.

In this paper, we propose Antigone, a principled approach that enables hyperparameter tuning for fairness without access to SA on training *and* validation data. Antigone enables effective hyperparameter tuning of methods like JTT (Liu et al., 2021), AFR (Qiu et al., 2023) (that currently assume ground-truth SAs on validation data) and improves fairness when GEORGE's and ARL's own hyperparameter tuning methods are

replaced with Antigone. Antigone works for fairness metrics including demographic parity, equal opportunity and worst sub-group accuracy.

Antigone starts with a baseline ERM model trained to predict target labels (note these target labels are known and different from SAs). Subgroups advantaged by the baseline ERM model will be over-represented in the set of correctly classified inputs; similarly, disadvantaged subgroups are over-represented in the set of incorrectly classified inputs. Hence, Antigone uses the ERM model's correctly and incorrectly classified validation data as proxies for advantaged and disadvantaged subgroups, respectively. We refer to these as pseudo-sensitive attributes (PSA). Note that PSA labels are *noisy*; some inputs from the advantaged subgroups will be incorrectly classified, and vice-versa. Can they still be used for hyperparameter tuning? Prior work shows that under certain theoretical assumptions on label noise, called the mutually corrupted (MC) noise model, fairness measured on PSA is proportional to ground-truth fairness (Lamy et al., 2019). Thus, PSA labels on validation data can still be used to compare different models for fairness, and consequently for hyperparameter tuning.

However, this sets up a new problem: how do we tune the hyperparameters of the ERM model itself to maximize the accuracy (or minimize the noise) of our PSA? We *cannot* directly measure PSA accuracy or noise since we do not have any ground-truth SA labels. Here, we show formally that under the MC noise model, minimizing noise is equivalent to maximizing the Euclidean distance between the mean (EDM) images in the correct and incorrect classes. Since the EDM *can* be directly measured, we train a family of ERM models with different hyper-parameters and pick the model with the largest EDM on the validation dataset. The PSAs obtained are then used to tune hyperparameters for fairness schemes like JTT, AFR, GEORGE and ARL.

For more intuition, consider the example in Figure 1 on the CelebA dataset. The target label is hair color, and the (unknown) SA is gender. The baseline ERM model discriminates against blond men. The correct set for the ground-truth blond class has only 4% blond men while the incorrect set has 65% blond men. This is also reflected in the mean images for these two classes: the correct set for the ground-truth blond individuals has more female features while the incorrect set has more male features. As noted above, Antigone picks an ERM model with the largest distance between these mean images so as to maximize PSA label accuracy.

We evaluate Antigone in conjunction with three state-of-art methods, JTT (Liu et al., 2021), AFR (Qiu et al., 2023), GEORGE (Sohoni et al., 2020) and ARL (Lahoti et al., 2020), on binary SA using demographic parity, equal opportunity, and worst subgroup accuracy as fairness metrics across the CelebA, Waterbirds and Adult datasets. Empirically, we find that: (1) Antigone produces more accurate PSA labels on validation data compared to GEORGE's unsupervised clustering approach (Table 1); (2) used with JTT (AFR), Antigone comes close to matching the fairness of JTT (AFR) tuned with ground-truth SA as shown in Table 2 (Table 5); and (3) improves the fairness of both GEORGE and ARL when Antigone's PSA labels are used instead of their own hyperparameter tuning methods (Table 3, Table 4). Specifically, the worst-group accuracy increases by 4.2% and 8.6% on the CelebA and Waterbirds datasets for GEORGE, respectively, and by up to 11.6% on the Adult dataset for ARL. Ablation and sensitivity studies demonstrate the effectiveness of Antigone's EDM metric versus alternatives (Table 1) and shed light on discrepancies between the ideal MC assumptions and its use in practice (Appendix Table 13). Overall, our key contributions are:

- We propose Antigone (subsection 2.2), a new method for the often overlooked problem of hyper-parameter tuning for fair classification in setting where sensitive attributes are unavailable on both training and validation data. Antigone generates PSA labels on validation data using the correctly and incorrectly classified examples of an ERM model as proxies for advantaged and disadvantaged subgroups.

- We propose an unsupervised approach to tune Antigone's own hyperparameters (specifically, those of its ERM model) by maximizing the Euclidean distance between the mean (EDM) images of the correctly and incorrectly classified sets. We theoretically justify this choice under the MC noise model, proving that maximizing EDM minimizes PSA label noise in an idealized setting (subsection 2.3). Empirically, we show that gap between our practical implementation and the idealized MC noise model is small.

- Experimentally, we find that Antigone based hyperparameter tuning boosts fairness for three state-of-art methods, JTT, AFR, GEORGE and ARL (section 3), even surpassing their own hyperparameter tuning techniques. Antigone also generates more accurate PSA labels compared to GEORGE's unsupervised clustering approach (section 4).

## 2 Proposed Methodology

We now describe Antigone, starting with the problem formulation (Section 2.1) followed by a description of the Antigone algorithm (Section 2.2).

### 2.1 Problem Setup

Consider a data distribution over set $\mathcal{D} = \mathcal{X} \times \mathcal{A} \times \mathcal{Y}$, the product of input data ($\mathcal{X}$), sensitive attributes ($\mathcal{A}$) and target labels ($\mathcal{Y}$) triplets. We are given a training set $D^{tr} = \{x_i^{tr}, a_i^{tr}, y_i^{tr}\}_{i=1}^{N^{tr}}$ with $N^{tr}$ training samples, and a validation set $D^{val} = \{x_i^{val}, a_i^{val}, y_i^{val}\}_{i=1}^{N^{val}}$ with $N^{val}$ validation samples. We will assume binary sensitive attributes ($\mathcal{A} \in \{0,1\}$) and target labels ($\mathcal{Y} \in \{0,1\}$).

We seek to train a machine learning model, say a deep neural network (DNN), which can be represented as a parameterized function $f_\theta : \mathcal{X} \to \mathcal{Y} \in \{0,1\}$, where $\theta \in \Theta$ are the trainable parameters, e.g., DNN weights and biases. Standard fairness unaware empirical risk minimization (ERM) optimizes over trainable parameters $\theta$ to minimize average binary cross-entropy loss on $D^{tr}$. Optimized model parameters $\theta^*$ are obtained by invoking a training algorithm, for instance stochastic gradient descent (SGD), on the training dataset and model, i.e., $\theta^{*,\gamma} = \mathcal{M}^{ERM}(D^{tr}, f_\theta, \gamma)$, where $\gamma \in \Gamma$ are hyper-parameters of the training algorithm including learning rate, training epochs etc. Hyper-parameters are tuned by evaluating models $f_{\theta^*,\gamma}$ for all $\gamma \in \Gamma$ on $D^{val}$ and picking the best model. More sophisticated algorithms like Bayesian optimization can also be used. Next, we review three commonly used fairness metrics that we account for in this paper.

**Demographic parity (DP):** DP requires the model's outcomes to be independent of sensitive attribute. In practice, we seek to minimize the demographic parity gap:

$$\Delta_\theta^{DP} = \mathbb{P}[f_\theta(X) = 1|A = 1] - \mathbb{P}[f_\theta(X) = 1|A = 0] \tag{1}$$

**Equal opportunity (EO):** EO aims to equalize only the model's true positive rates across sensitive attributes. In practice, we seek to minimize

$$\Delta_\theta^{EO} = \mathbb{P}[f_\theta(X) = 1|A = 1, Y = 1] - \mathbb{P}[f_\theta(X) = 1|A = 0, Y = 1] \tag{2}$$

**Worst-group accuracy (WGA):** WGA seeks to maximize the minimum accuracy over all sub-groups (over sensitive attributes and target labels). That is, we seek to maximize:

$$WGA_\theta = \min_{a \in \{0,1\}, y \in \{0,1\}} \mathbb{P}[f(x) = y|A = a, Y = y] \tag{3}$$

In all three settings, we seek to train models that optimize fairness under a constraint on average *target label accuracy*, *i.e.,* accuracy in predicting the target label. For example, for equal opportunity, we seek $\theta^* = \arg\min_{\theta \in \Theta} \Delta_\theta^{EO}$ such that $\mathbb{P}[f_\theta(x) = Y] \in [Acc_{lower}^{thr}, Acc_{upper}^{thr})$, where $Acc_{lower}^{thr}$ and $Acc_{upper}^{thr}$ are user-specified lower and upper bounds on target label accuracies, respectively.

### 2.2 Antigone Algorithm

We now describe the Antigone algorithm which consists of three main steps. In step 1, we train multiple ERM models that each provide pseudo sensitive attribute (PSA) labels on validation data. In step 2, we use the proposed EDM metric to pick a single ERM model from step 1 with the most accurate PSA labels. Finally, in step 3, we use the PSA labelled validation set from step 2 to tune the hyper-parameters of methods like JTT that train fair classifiers without SA labels on training data.

**Step 1: Generating PSA labels on validation set.** In step 1, we use the training dataset and standard ERM training to obtain a set of classifiers, $\theta^{*,\gamma} = \mathcal{M}^{ERM}(D^{tr}, f_\theta, \gamma)$, each corresponding to a different value of training hyper-parameters $\gamma \in \Gamma$. As we discuss in Section 2.1, these include learning rate, weight decay and number of training epochs. Each classifier, which predicts the target label for a given input, generates a validation set with PSA labels as follows:

$$D_{\mathrm{PSA}}^{val,\gamma} = \{x_i^{val}, a_i^{val,\gamma}, y_i^{val}\}_{i=1}^{N^{val}} \quad \forall \gamma \in \Gamma, \text{where} \tag{4}$$

$$a_i^{val,\gamma} = \begin{cases} 1, & \text{if } f_{\theta^{*,\gamma}}(x_i^{val}) = y_i^{val} \\ 0, & \text{otherwise.} \end{cases} \tag{5}$$

where $a_i^{val,\gamma}$ now refers to PSA labels. In the next step, we search over the set $\Gamma$ to find hyperparameters $\gamma \in \Gamma$ that maximize PSA accuracy.

**Step 2: Picking the most accurate PSA labeller.** From Step 1, let the correct set be $X_{A=1,\mathrm{PSA}}^{val,\gamma} = \{x_i^{val} : a_i^{val,\gamma} = 1\}$ and the incorrect set be $X_{A=0,\mathrm{PSA}}^{val,\gamma} = \{x_i^{val} : a_i^{val,\gamma} = 0\}$. We define Euclidean distance between the means (EDM) of these sets as:

$$EDM^\gamma = \|\mu(X_{A=1,\mathrm{PSA}}^{val,\gamma}) - \mu(X_{A=0,\mathrm{PSA}}^{val,\gamma})\|_2, \tag{6}$$

where $\mu(.)$ represents the empirical mean of a dataset. Antigone picks $\gamma^*$ that maximizes EDM, i.e., $\gamma^* = \arg\max_{\gamma \in \Gamma} EDM^\gamma$. We justify this choice in two ways. Intuitively, PSA labels on validation data distinguish between advantaged and disadvantaged classes, e.g., placing blond men and blond women in different groups as in Figure 1, resulting in larger differences in the mean images of the two groups. Formally, we show the optimality of this strategy under the MC noise model in Subsection 2.3.

**Step 3: Training a fair model.** Step 2 yields $D_{\mathrm{PSA}}^{val,\gamma^*}$, a validation dataset with (estimated) pseudo sensitive attribute labels. We can provide $D_{\mathrm{PSA}}^{val,\gamma^*}$ as an input to any method that trains fair models without access to SA on training data, but requires a validation set with SA labels to tune its own hyper-parameters. In our experimental results, we use $D_{\mathrm{PSA}}^{val,\gamma^*}$ to tune the hyper-parameters of JTT (Liu et al., 2021), GEORGE (Sohoni et al., 2020) and ARL (Lahoti et al., 2020).

## 2.3 Analyzing Antigone under Ideal MC Noise

Prior theoretical work (Lamy et al., 2019) has studied the impact of noisy sensitive attribute labels on fairness under the "mutually contaminated" (MC) noise model (Scott et al., 2013). Here, it is assumed that we have access to PSA labels, $X_{A=0,\mathrm{PSA}} \in \mathcal{X}$ and $X_{A=1,\mathrm{PSA}} \in \mathcal{X}$, corresponding to disadvantaged (PSA = 0) and advantaged (PSA = 1) groups, respectively, that are contaminated (or, noisy) versions of their corresponding ground-truth SA labels, $X_{A=0} \in \mathcal{X}$ and $X_{A=1} \in \mathcal{X}$. Specifically, $P(X|\mathrm{PSA} = 1) = (1-\alpha)P(X|A=1) + \alpha P(X|A=0)$ and $P(X|\mathrm{PSA} = 0) = \beta P(X|A=1) + (1-\beta)P(X|A=0)$ decompositions are satisfied under the MC noise model, but we follow the notation described in prior work (Scott et al., 2013; Lamy et al., 2019) to denote the MC noise model in our setting as:

$$X_{A=1,\mathrm{PSA}} = (1-\alpha)X_{A=1} + \alpha X_{A=0} \text{ and } X_{A=0,\mathrm{PSA}} = \beta X_{A=1} + (1-\beta)X_{A=0} \tag{7}$$

where $\alpha$ and $\beta$ are noise parameters. With some abuse of terminology for conciseness, Equation 7 says that fraction $\alpha$ of the pseudo advantaged group, $X_{A=1,\mathrm{PSA}}$, is contaminated with data from the disadvantaged group, and fraction $\beta$ of the pseudo disadvantaged group, $X_{A=0,\mathrm{PSA}}$, is contaminated with data from the advantaged group. We construct $D_{A=0,\mathrm{PSA}}$ by appending input instances in $X_{A=0,\mathrm{PSA}}$ with their corresponding PSA labels (*i.e.,* $a_i = 0$) and target labels, respectively. We do the same for $D_{A=1,\mathrm{PSA}}$. The noise can also be target dependent, in which case we use $\alpha_i$ and $\beta_i$ as noise parameters for label $i$. Under this model, Lamy et al. (2019) show the following result:

**Proposition 2.1.** *(Lamy et al., 2019) Under the ideal MC noise model in Equation 7, DP and EO gaps measured on the noisy datasets are proportional to the true DP and EO gaps. Mathematically:*

$$\Delta^{DP}(D_{A=0,PSA} \cup D_{A=1,PSA}) = (1 - \alpha - \beta)\Delta^{DP}(D_{A=0} \cup D_{A=1}), and \tag{8}$$

$$\Delta^{EO}(D_{A=0,PSA} \cup D_{A=1,PSA}) = (1 - \alpha_1 - \beta_1)\Delta^{EO}(D_{A=0} \cup D_{A=1}). \qquad (9)$$

Equation 8 and Equation 9 show that under the ideal MC noise model, the DP and EO gaps can be minimized using PSA labels instead of ground-truth SA labels, although asymptotically with infinite validation data samples. In practice, we seek to minimize the total noise $\alpha + \beta$, or equivalently maximize the proportionality constant $1 - \alpha - \beta$ to obtain the most reliable fairness estimates. We show that this can be done by maximizing EDM.

**Lemma 2.2.** *Assume $X_{A=0,PSA}$ and $X_{A=1,PSA}$ correspond to the input data of noisy datasets in the ideal MC model. Then, maximizing the EDM between $X_{A=0,PSA}$ and $X_{A=1,PSA}$, i.e., $\|\mu(X_{A=0,PSA}) - \mu(X_{A=1,PSA})\|_2$ maximizes $1 - \alpha - \beta$.*

*Proof.* From Equation 7, we can see that $\|\mu(X_{A=0,\mathrm{PSA}}) - \mu(X_{A=1,\mathrm{PSA}})\|_2 = (1 - \alpha - \beta)^2\|\mu(X_{A=0}) - \mu(X_{A=1})\|_2$. Here $\|\mu(X_{A=0}) - \mu(X_{A=1})\|_2$ is the EDM between the ground truth advantaged and disadvantaged data and is therefore a constant. Hence, maximizing EDM between $X_{A=0,\mathrm{PSA}}$ and $X_{A=1,\mathrm{PSA}}$ maximizes $1 - \alpha - \beta$. $\qquad\qquad\square$

*Remark* 2.3. The MC noise model assumes *independent* label noise. However, when using ERM classifiers to generate PSAs, this noise can be instance dependent. Although we use the simplified MC noise model to inform our practical implementation, we do not claim that Antigone inherits the MC model's theoretical guarantees. In Table 13, we do show empirically that the gap between fairness achieved with Antigone and under ideal MC noise is small.

## 3 Experimental Setup

### 3.1 Baseline Methods

We evaluate Antigone with state-of-the-art fairness methods that work without SA on training data: JTT (Liu et al., 2021), GEORGE Sohoni et al. (2020), ARL Lahoti et al. (2020), and AFR (Qiu et al., 2023). GEORGE and ARL additionally do not require SA on validation data. Below, we describe these baselines.

**JTT:** JTT operates in two stages. In the first stage, a biased model is trained using $T$ epochs of standard ERM training to identify the incorrectly classified training examples. In the second stage, the misclassified examples are upsampled $\lambda$ times, and the model is trained again to completion with standard ERM. The hyperparameters of stage 1 and stage 2 classifiers, including early stopping epoch $T$, learning rate and weight decay for stage 1 and upsampling factor $\lambda$ for stage 2, are jointly tuned using a validation dataset with ground-truth SA labels. We refer to this as the **Ground-Truth + JTT** baseline.

**GEORGE:** GEORGE is a competing approach to Antigone in that it does not assume access to SA on either training or validation data. GEORGE operates in two stages: In stage 1, an ERM model is trained until completion on the ground-truth target labels. The activation in the penultimate layer of the ERM model are clustered into $k$ clusters to generate PSA labels on both the training and validation datasets. In Stage 2, these PSA are used to train a Group DRO model Sagawa* et al. (2020) and tune its early stopping hyper-parameter.

**ARL:** Adversarially Reweighted Learning (ARL) seeks to improve the worst-group performance without access to SA on either training or validation datasets. ARL framework considers a mini-max game between a learner and adversary. The goal of the learner is to output fair predictions by minimizing the weighted cross entropy classification loss function. Whereas, the adversary maximizes the weighted cross entropy loss so as to identify high loss training data points and upweight them during the training of the learner. Since ARL does not assume access to SA on validation set, it tries to maximize the target label accuracy by performing a grid search over the joint hyper-parameter space of both learner and adversary.

**AFR:** Automatic Feature Reweighting is a lightweight framework that operates in two stages. In the first stage, a standard ERM model is trained on 80% of the training dataset to minimize the ERM objective. In the second stage, AFR retrains only the last layer of the ERM model using the remaining 20% of the training dataset with a weighted loss function. The weights of each sample, $i$, are determined by:

$$w_i = \frac{\kappa_{y_i}\exp(-\gamma\hat{p}_i)}{\Sigma_{j=1}^{M}\kappa_{y_j}\exp(-\gamma\hat{p}_j)} \tag{10}$$

These weights emphasize the samples where the standard ERM model from stage 1 underperforms. The hyper-parameter, $\gamma$, is used to control how much to up-weight examples with poor ground-truth target label prediction probability ($\hat{p}$), ensuring that examples from the disadvantaged group are given greater importance in stage 2. Additionally, AFR balances the target class labels using $\kappa_y$, which is the reciprocal of the number of examples belonging to class $y$ in the re-training set. In stage 2, AFR also uses a regularization term to prevent the last layer from focusing only on minority examples, potentially at the cost of degrading performance on majority group examples to an unacceptable level. The strength of this regularization is controlled by the hyper-parameter $\tau$. The hyper-parameters of the second stage are tuned using a validation dataset with ground-truth SA labels. We refer to this as the **Ground-Truth + AFR** baseline.

## 3.2 Antigone + Baseline

**Antigone+JTT:** Here, we replace the ground-truth SA in the validation dataset with PSA obtained from Antigone and use it to tune JTT's hyper-parameters.

**Antigone+GEORGE:** For a fair comparison with GEORGE, we replace its stage 1 with Antigone, and use the resulting validation PSA labels to tune the hyper-parameters of GEORGE's stage 2.

**Antigone+ARL:** Instead of tuning the target label accuracy on the validation set, we use Antigone's validation PSA labels to tune the hyper-parameters of ARL. We also tune the hyper-parameters of ARL with ground-truth SA labels and refer to it as **Ground-Truth+ARL**.

**Antigone+AFR:** Here, our primary goal is to improve WGA, as AFR seeks to mitigate spurious correlations by improving WGA with the assumption of having access to ground-truth SA on the validation dataset. We substitute the ground-truth SA in the validation dataset with PSA acquired from Antigone and use it to tune AFR's hyperparameters.

## 3.3 Datasets and Parameter Settings

We empirically evaluate Antigone on the CelebA and Waterbirds datasets, which allow for a direct comparison with related work (Liu et al., 2021; Sohoni et al., 2020). We also evaluate Antigone on UCI Adult Dataset, a tabular dataset commonly used in the fairness literature to directly compare with ARL Lahoti et al. (2020) (see Appendix A) for more details.

**CelebA Dataset:** CelebA (Liu et al., 2015) is an image dataset, consisting of 202,599 celebrity face images annotated with 40 attributes including gender, hair colour, age, smiling, etc. The task is to predict hair color, which is either blond $Y = 1$ or non-blond $Y = 0$ and the sensitive attribute is gender $A = \{\text{Men}, \text{Women}\}$. In all our experiments using CelebA dataset, we fine-tune a pre-trained ResNet50 architecture for a total of 50 epochs using SGD optimizer and a batch size of 128. We tune JTT over the same hyper-parameters as in their paper: three pairs of learning rates and weight decays, $(1e-04, 1e-04), (1e-04, 1e-02), (1e-05, 1e-01)$ for both stages, and over ten early stopping points up to $T = 50$ and $\lambda \in \{20, 50, 100\}$ for stage 2. For Antigone, we explore over the same learning rate and weight decay values, as well as early stopping at any of the 50 training epochs. We report results for DP, EO and WGA fairness metrics. In each case, we seek to optimize fairness while constraining average target label accuracy to ranges $\{[90, 91), [91, 92), [92, 93), [93, 94), [94, 95)\}$.

**Waterbirds Dataset:** Waterbirds is a synthetically generated dataset, containing 11,788 images of water and land birds overlaid on top of either water or land backgrounds (Sagawa* et al., 2020). The task is to predict the bird type, which is either a waterbird $Y = 1$ or a landbird $Y = 0$ and the sensitive attribute is the background $A = \{\text{Water background}, \text{Land background}\}$. In all our experiments using Waterbirds dataset, we fine-tune ResNet50 architecture for a total of 300 epoch using the SGD optimizer and a batch size of 64. We tune JTT over the same hyper-parameters as in their paper: three pairs of learning rates and weight decays, $(1e - 03, 1e - 04), (1e - 04, 1e - 01), (1e - 05, 1.0)$ for both stages, and over 14 early stopping points up to $T = 300$ and $\lambda \in \{20, 50, 100\}$ for stage 2. For Antigone, we explore over the same learning rate and weight decay values, as well as early stopping points at any of the 300 training epochs. In each case, we seek to optimize fairness while constraining average accuracy to ranges $\{[94, 94.5), [94.5, 95), [95, 95.5), [95.5, 96), [96, 96.5)\}$.

In case of GEORGE, for both CelebA and Waterbirds datasets, we use the same architecture and early stopping stage 2 hyper-parameters ($T = 50$ for CelebA and $T = 300$ for Waterbirds) reported in their paper. For Antigone+GEORGE, we replace GEORGE's stage 1 with the Antigone model, which is identified by searching over the same hyperparameter space as in Antigone+JTT.

In case of AFR, for both CelebA and Waterbirds, we fine-tune a pre-trained ResNet50 architecture and tune its hyper-parameters to minimize the cross-entropy loss function in stage 1. In stage 2, we re-train the last layer by searching over the hyper-parameters space reported in their paper. Specifically, for CelebA, we tune over early stopping stage 2 epochs ($T = 1000$), $\gamma$ from 10 points linearly spaced between $[1, 3]$, learning rate $= 2e - 2$, and $\tau \in \{0.001, 0.01, 0.1\}$. For Waterbirds, we tune over early stopping stage 2 epochs ($T = 500$), $\gamma$ from 33 points linearly spaced between $[4, 20]$, learning rate $= 1e - 2$, and $\tau \in \{0, 0.1, 0.2, 0.3, 0.4\}$. For Antigone+AFR, we tune AFR's stage 2 hyper-parameters using Antigone's PSA, which are identified by searching over the same hyperparameter space as in Antigone+JTT.

## 4 Experimental Results

**Accuracy of Antigone's PSA labels:** Antigone seeks to generate accurate PSA labels on validation data, referred to as *pseudo label accuracy*, based on the EDM criterion (Lemma 2.2). In Figure 2, we empirically validate Lemma 2.2 by plotting EDM and noise parameters $\alpha_1$ (contamination in advantaged group), $\beta_1$ (contamination in disadvantaged group) and $1 - \alpha_1 - \beta_1$ (proportionality constant between true and estimated fairness) on Waterbirds dataset (similar plot for CelebA dataset is in Appendix Figure 3(b)). From the figure, we observe that in both cases the EDM metric indeed captures the trend in $1 - \alpha_1 - \beta_1$, enabling early stopping at an epoch that minimizes contamination. The best early stopping points based on EDM and oracular knowledge of $1 - \alpha_1 - \beta_1$ are shown in a blue dot and star, respectively, and are very close.

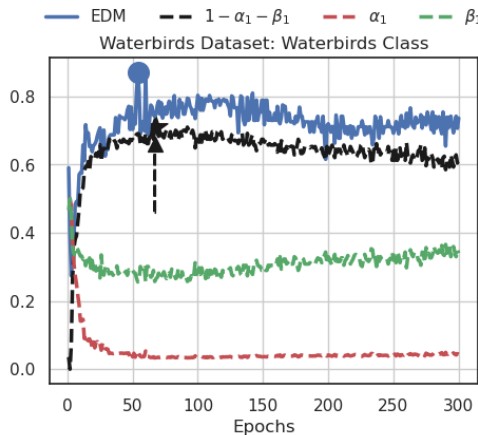

Figure 2: Euclidean Distance between Means (EDM) and noise parameters ($\alpha_1, \beta_1$ and $1 - \alpha_1 - \beta_1$) for the positive target class of Waterbirds dataset. Blue dot indicates the model picked by Antigone, while black star indicates the model that maximizes $1 - \alpha_1 - \beta_1$.

In Table 1 we compare Antigone's PSA labels' F1 Score to GEORGE with the baseline $k = 5$ and with $k = 2$ clusters on CelebA and Waterbirds. We find that Antigone outperforms GEORGE on all but one sub-group in CelebA, and all Waterbirds. Table 1 also reports results on a version of Antigone that uses standard ERM training instead of EDM (Antigone (w/o EDM)). We find that Antigone provides higher pseudo-label accuracy compared to this baseline. Appendix Table 6 shows precision and recall of Antigone's PSA labels and reaches the same conclusion. We also study the impact of varying the fraction of disadvantaged group individuals from 5%, 20%, 35%, 50% in the CelebA dataset (see Appendix Table 7). As the dataset gets more balanced, the models themselves are more fairer, and PSA label accuracy reduces (as expected). Nonetheless, disadvantaged group individuals are over-represented in incorrect sets for up to 35% imbalance.

Table 1: F1 scores and *pseudo label accuracies* (Ps. Acc.) We mark the best performance in bold. BM (blond men), BW (blond women), NBW (non-blond women) and NBM (non-blond men) for CelebA; WL (waterbirds landbkgd), WW (waterbirds waterbkgd), LW (landbirds waterbkgd) and LL (landbirds landbkgd) for Waterbirds.

| | Antigone (w/o EDM) | GEORGE | GEORGE ($k = 2$) | Antigone (w/ EDM) |
|---|---|---|---|---|
| | | CelebA (F1 Scores) | | |
| BM | $0.28_{\pm 0.01}$ | $0.13_{\pm 0.02}$ | $0.12_{\pm 0.01}$ | $\mathbf{0.35}_{\pm \mathbf{0.04}}$ |
| BW | $0.95_{\pm 0.01}$ | $0.43_{\pm 0.04}$ | $0.51_{\pm 0.02}$ | $\mathbf{0.96}_{\pm \mathbf{0.00}}$ |
| NBW | $0.22_{\pm 0.02}$ | $0.42_{\pm 0.01}$ | $\mathbf{0.6}_{\pm \mathbf{0.01}}$ | $0.22_{\pm 0.01}$ |
| NBM | $0.67_{\pm 0.01}$ | $0.4_{\pm 0.02}$ | $0.31_{\pm 0.01}$ | $\mathbf{0.68}_{\pm \mathbf{0.01}}$ |
| Ps. Acc. | $0.59_{\pm 0.01}$ | $0.33_{\pm 0.01}$ | $0.48_{\pm 0.00}$ | $\mathbf{0.60}_{\pm \mathbf{0.00}}$ |
| | | Waterbirds (F1 Scores) | | |
| WL | $0.41_{\pm 0.02}$ | $0.43_{\pm 0.02}$ | $0.52_{\pm 0.01}$ | $\mathbf{0.76}_{\pm \mathbf{0.03}}$ |
| WW | $0.72_{\pm 0.00}$ | $0.36_{\pm 0.02}$ | $0.43_{\pm 0.02}$ | $\mathbf{0.83}_{\pm \mathbf{0.01}}$ |
| LW | $0.58_{\pm 0.02}$ | $0.44_{\pm 0.03}$ | $0.55_{\pm 0.03}$ | $\mathbf{0.78}_{\pm \mathbf{0.04}}$ |
| LL | $0.76_{\pm 0.01}$ | $0.34_{\pm 0.02}$ | $0.55_{\pm 0.03}$ | $\mathbf{0.84}_{\pm \mathbf{0.02}}$ |
| Ps. Acc. | $0.68_{\pm 0.01}$ | $0.30_{\pm 0.02}$ | $0.53_{\pm 0.03}$ | $\mathbf{0.81}_{\pm \mathbf{0.02}}$ |

**Antigone+JTT:** In Table 2, we compare the test target label accuracy and fairness achieved by Antigone with JTT (Antigone+JTT) vs. JTT using ground-truth SA (Ground-Truth+JTT). On DP and EO, Antigone+JTT is very close to Ground-Truth+JTT in terms of both target label accuracy and fairness, and substantially improves on standard ERM. Antigone+JTT improves WGA from 38.7% for standard ERM to 68.1% at the expense of 3% target label accuracy drop. Ground-Truth+JTT improves WGA further up to 78.6% but with a 4.4% target label accuracy drop. Waterbirds (Appendix Table 11) and UCI Adult (Appendix Table 12) have the same trends.

**Comparison with GEORGE:** As already noted in Table 1, Antigone's PSA labels are more accurate and have higher F1 scores than GEORGE's. In Table 3, Antigone+GEORGE shows marked improvements over GEORGE in both WGA (8.6% higher) and target label accuracy on Waterbirds. For CelebA, Antigone+GEORGE has a 4.2 % higher WGA but with a small drop of 0.4% in target label accuracy. Fairness improvement are statistically significant under paired t-tests; over 10 runs, Antigone+GEORGE always equals or betters GEORGE in terms of WGA (Appendix Figure 5).

**Comparison with ARL:** In Table 4 (full table in Appendix Table 14), we compare the target label accuracies and fairness of Antigone+ARL vs. just ARL for different validation accuracy thresholds on the Adult UCI. Antigone+ARL has 4.5%-11.6% higher WGA than ARL alone, but with a small ($< 0.6\%$) drop in target label accuracy. Similar observations hold on DP Gap and EO Gap.

**Comparison with AFR:** In Table 5, we compare the test target label accuracies and WGA achieved by Antigone with AFR (Antigone+AFR) vs. AFR using ground-truth SA (Ground-Truth+AFR). Antigone + AFR is very close to Ground-Truth + AFR in terms of both target label accuracy and WGA, and substantially improves on standard ERM. On CelebA, Antigone + AFR improves WGA from 41% for standard ERM to

Table 2: (Avg. target label accuracy, Fairness) on test data for different validation accuracy thresholds on the CelebA dataset. Lower DP and EO gaps are better. Higher WGA is better.

| Val. Thresh. | Method | DP Gap | EO Gap | Worst-group Acc. |
|---|---|---|---|---|
| [94, 95) | Antigone + JTT | $(94.6, 15.0)_{\pm\ (0.2,\ 0.7)}$ | $(94.7, 30.1)_{\pm\ (0.2,\ 3.2)}$ | $(94.4, 59)_{\pm\ (0.2,\ 4.7)}$ |
| | Ground-Truth + JTT | $(94.7, 14.9)_{\pm\ (0.2,\ 0.6)}$ | $(94.5, 30.4)_{\pm\ (0.2,\ 2.3)}$ | $(94.3, 62.1)_{\pm\ (0.3,\ 3.2)}$ |
| [93, 94) | Antigone + JTT | $(93.7, 13.1)_{\pm\ (0.2,\ 0.7)}$ | $(93.6, 26.4)_{\pm\ (0.4,\ 5.0)}$ | $(93.4, 62.6)_{\pm\ (0.2,\ 7.0)}$ |
| | Ground-Truth + JTT | $(93.6, 13.1)_{\pm\ (0.1,\ 0.6)}$ | $(93.6, 22.7)_{\pm\ (0.3,\ 2.7)}$ | $(93.4, 67.9)_{\pm\ (0.1,\ 1.9)}$ |
| [92, 93) | Antigone + JTT | $(92.7, 11.1)_{\pm\ (0.2,\ 0.5)}$ | $(92.3, 20.2)_{\pm\ (0.2,\ 3.4)}$ | $(92.7, 68.1)_{\pm\ (0.4,\ 3.7)}$ |
| | Ground-Truth + JTT | $(92.7, 11.2)_{\pm\ (0.3,\ 0.5)}$ | $(92.7, 16.9)_{\pm\ (0.4,\ 2.9)}$ | $(92.7, 72.5)_{\pm\ (0.2,\ 1.3)}$ |
| [91, 92) | Antigone + JTT | $(91.7, 9.6)_{\pm\ (0.1,\ 0.5)}$ | $(91.5, 16.3)_{\pm\ (0.3,\ 3.4)}$ | $(91.3, 63.2)_{\pm\ (0.3,\ 2.6)}$ |
| | Ground-Truth + JTT | $(91.8, 9.7)_{\pm\ (0.2,\ 0.5)}$ | $(91.8, 10.1)_{\pm\ (0.3,\ 4.1)}$ | $(91.8, 77.3)_{\pm\ (0.1,\ 2.4)}$ |
| [90, 91) | Antigone + JTT | $(91.0, 8.3)_{\pm\ (0.2,\ 0.4)}$ | $(90.9, 13.1)_{\pm\ (0.1,\ 3.6)}$ | $(90.9, 63.1)_{\pm\ (0.5,\ 4.4)}$ |
| | Ground-Truth + JTT | $(91.0, 8.4)_{\pm\ (0.2,\ 0.4)}$ | $(90.7, 6.8)_{\pm\ (0.4,\ 3.7)}$ | $(91.4, 78.6)_{\pm\ (0.2,\ 2.0)}$ |
| | ERM | $(95.8, 18.6)_{\pm\ (0.0,\ 0.3)}$ | $(95.8, 46.4)_{\pm\ (0.0,\ 2.2)}$ | $(95.8, 38.7)_{\pm\ (0.0,\ 2.8)}$ |

Table 3: Performance of GEORGE using Antigone's validation PSA compared with GEORGE by itself. We observe that on CelebA and Waterbirds dataset, Antigone + GEORGE out-performs GEORGE, even if GEORGE assumes knowledge of number of clusters ($k = 2$) in its clustering step. $*$ ($**$) indicates $p-\text{value} < 0.01(0.001)$.

| | CelebA | | Waterbirds | |
|---|---|---|---|---|
| Method | Avg Acc | WGA | Avg Acc | WGA |
| ERM | **95.75** | 35.14 | 95.91 | 29.70 |
| GEORGE | 93.61 | 60.44 | 95.39 | 49.15 |
| Antigone + GEORGE | 93.56 | 62.45* | **95.99**$^{**}$ | **57.73**$^{**}$ |
| GEORGE (k=2) | 94.62 | 60.75 | 94.61 | 42.98 |
| Antigone + GEORGE (k=2) | 94.18 | **64.94**$^{**}$ | 95.56$^{**}$ | 52.80* |

Table 4: Comparison of (Avg. target label accuracy, Fairness) between ARL using Antigone's noisy validation data, ARL alone, and ground-truth validation data on the UCI Adult dataset. Lower DP and EO gaps indicate better fairness, while higher WGA is better. Antigone + ARL consistently outperforms ARL across various validation accuracy thresholds and fairness metrics. The $p$-values are marked with $*$ accordingly: $*$ for $p < 0.1$, $**$ for $p < 0.05$, and $***$ for $p < 0.01$.

| Val. Thresh. | Method | DP Gap | EO Gap | Worst-group Acc. |
|---|---|---|---|---|
| [84.5, 85) | Antigone + ARL | $(\textbf{84.51, 16.84})^{**}\ _{\pm\ (0.13,\ 1.08)}$ | $(\textbf{84.1, 5.64})^{***}\ _{\pm\ (0.08,\ 1.43)}$ | $(\textbf{84.14, 59.68})^{***}\ _{\pm\ (0.13,\ 1.27)}$ |
| | ARL | $(84.53, 19.01)\ _{\pm\ (0.13,\ 0.97)}$ | $(84.53, 9.35)\ _{\pm\ (0.13,\ 0.88)}$ | $(84.53, 55.08)\ _{\pm\ (0.13,\ 3.03)}$ |
| | Ground-Truth + ARL | $(84.5, 15.86)\ _{\pm\ (0.24,\ 1.02)}$ | $(84.43, 4.97)\ _{\pm\ (0.17,\ 1.23)}$ | $(84.08, 62.66)\ _{\pm\ (0.32,\ 0.71)}$ |
| [84, 84.5) | Antigone + ARL | $(\textbf{84.12, 15.55})^{***}\ _{\pm\ (0.21,\ 1.14)}$ | $(\textbf{83.7, 6.65})^{**}\ _{\pm\ (0.22,\ 1.61)}$ | $(\textbf{83.47, 60.9})^{**}\ _{\pm\ (0.22,\ 2.1)}$ |
| | ARL | $(84.02, 18.35)\ _{\pm\ (0.17,\ 1.04)}$ | $(84.02, 8.04)\ _{\pm\ (0.17,\ 1.21)}$ | $(84.02, 55.69)\ _{\pm\ (0.17,\ 2.93)}$ |
| | Ground-Truth + ARL | $(84.23, 15.26)\ _{\pm\ (0.12,\ 0.9)}$ | $(83.81, 5.57)\ _{\pm\ (0.12,\ 1.58)}$ | $(83.62, 64.74)\ _{\pm\ (0.14,\ 0.65)}$ |
| [83.5, 84) | Antigone + ARL | $(\textbf{83.29, 15.13})^{***}\ _{\pm\ (0.21,\ 1.17)}$ | $(\textbf{83.07, 5.36})^{**}\ _{\pm\ (0.22,\ 2.86)}$ | $(\textbf{82.93, 62.17})^{***}\ _{\pm\ (0.13,\ 1.63)}$ |
| | ARL | $(83.54, 19.43)\ _{\pm\ (0.12,\ 1.17)}$ | $(83.54, 10.52)\ _{\pm\ (0.12,\ 2.16)}$ | $(83.54, 53.72)\ _{\pm\ (0.12,\ 2.66)}$ |
| | Ground-Truth + ARL | $(83.53, 14.7)\ _{\pm\ (0.19,\ 0.87)}$ | $(83.17, 4.34)\ _{\pm\ (0.17,\ 0.69)}$ | $(83.31, 66.61)\ _{\pm\ (0.24,\ 1.6)}$ |
| [83, 83.5) | Antigone + ARL | $(\textbf{82.98, 15.05})^{***}\ _{\pm\ (0.18,\ 1.45)}$ | $(\textbf{82.69, 5.4})^{*}\ _{\pm\ (0.16,\ 3.15)}$ | $(\textbf{82.55, 65.85})^{***}\ _{\pm\ (0.31,\ 0.58)}$ |
| | ARL | $(83.2, 18.24)\ _{\pm\ (0.13,\ 2.45)}$ | $(83.2, 8.69)\ _{\pm\ (0.13,\ 3.68)}$ | $(83.2, 54.21)\ _{\pm\ (0.13,\ 4.99)}$ |
| | Ground-Truth + ARL | $(82.86, 14.84)\ _{\pm\ (0.13,\ 1.26)}$ | $(83.04, 6.03)\ _{\pm\ (0.13,\ 1.56)}$ | $(82.47, 66.75)\ _{\pm\ (0.16,\ 1.56)}$ |
| | ERM | $(84.69, 18.27)\ _{\pm\ (0.08,\ 0.5)}$ | $(84.69, 9.39)\ _{\pm\ (0.08,\ 1.11)}$ | $(84.69, 53.36)\ _{\pm\ (0.08,\ 1.97)}$ |

81% at the expense of 5% target label accuracy drop. Ground-Truth + AFR improves WGA further up to 82% with a 4% target label accuracy drop. Similar observations hold for the Waterbirds dataset also.

Table 5: Performance of AFR using Antigone's validation PSA compared with AFR by itself. Higher WGA is better. We observe that Antigone + AFR is close to the performance of Ground-Truth + AFR.

| Method | CelebA | | Waterbirds | |
|---|---|---|---|---|
| | Avg Acc | WGA | Avg Acc | WGA |
| ERM | $0.96_{\pm 0.01}$ | $0.41_{\pm 0.01}$ | $0.98_{\pm 0.02}$ | $0.64_{\pm 0.01}$ |
| Antigone + AFR | $0.91_{\pm 0.01}$ | $0.81_{\pm 0.01}$ | $0.92_{\pm 0.04}$ | $0.82_{\pm 0.02}$ |
| Ground-Truth + AFR | $0.92_{\pm 0.00}$ | $0.82_{\pm 0.01}$ | $0.93_{\pm 0.03}$ | $0.84_{\pm 0.02}$ |

**Ablation Studies:** We perform two ablation experiments to understand the benefits of Antigone's proposed EDM metric. We already noted in Table 1 that Antigone with the proposed EDM metric produces higher quality PSA labels compared to a version of Antigone that picks hyper-parameters using standard ERM. We evaluated these two approaches using JTT's training algorithm and find that Antigone with EDM results in a 5.7% increase in WGA and a small 0.06% increase in average target label accuracy. Second, in Appendix Table 13, we also compare Antigone+JTT against a synthetically labeled validation dataset that exactly follows the ideal MC noise model in Section 2.3. We find that on DP Gap and EO Gap fairness metrics, Antigone 's results are comparable (in fact sometimes slightly better) with those derived from the ideal MC model. On WGA, the most challenging fairness metric to optimize for, we find that the ideal MC model has a best-case WGA of 73.9% compared to Antigone's 66.7%. This reflects the loss in fairness due to the gap between the assumptions of the idealized model versus Antigone's implementation; however, the reduction in fairness is marginal when compared to the ERM baseline which has only a 38% WGA.

**Applicability to backbone models** We conducted additional experiments to evaluate Antigone's performance using large pre-trained models. Specifically, we fine-tuned a pre-trained state-of-art large transformer model, ViT-B/16 (Dosovitskiy et al., 2020), on the Waterbirds dataset, resulting in target label accuracy of 99% and WGA of 82% for the ERM model (refer to Appendix B.1 for more details). In Appendix Table 8, we compare Antigone-ViT-B/16's PSA labels to GEORGE with k = 5 and k = 2 clusters. We find that Antigone-ViT-B/16 outperforms GEORGE. Furthermore, we compare the test target label accuracies and WGA achieved by Antigone-ViT-B/16 + AFR vs. Ground-Truth+AFR (Appendix Table 10). The results are consistent with prior observations and show that Antigone-ViT-B/16 can be successfully used with large pre-trained models.

## 5    Limitations

Antigone has some notable limitations that we discuss here, along with potential avenues to mitigate these concerns. First, in its current form, Antigone only explicitly deals with *binary* sensitive attributes. In practice, multiple subgroups could in fact be over-represented in the incorrect set, and as such accounted for during hyperparameter tuning but *not* explicitly. We note that downstream robustness methods, like JTT, that we demonstrate Antigone in conjunction with and others like Creager et al. (2021) have the same limitation. Antigone+JTT can improve fairness for these sub-groups as a whole but cannot, for example, have different up-weighting factors for each subgroup. However, this limitation is not fundamental and can be addressed by further sub-dividing the incorrect set, or via multiple rounds of Antigone+JTT where in each round we address any remaining fairness gaps. We note also that although GEORGE addresses multiple (k) subgroups, tuning k is challenging and results for larger k are sometimes worse than k = 2.

A second concern is whether the assumptions of the MC noise model, independent label noise in particular, hold strictly. While they do not, we are not claiming the theoretical fairness guarantees of Lamy et al. (2019). Antigone is a practical solution to improve fairness like other works we evaluate against. In Appendix Table 13, we do compare Antigone+JTT against a synthetically labeled validation dataset that exactly follows the MC noise model and find that Antigone's reduction in fairness is marginal. In Figure 2,

we also empirically validate that the proportionality constant $1 - \alpha - \beta$ minimizes the gap between the true and estimated fairness values.

## 6  Related Work

Methods that seek to achieve fairness are of three types: pre-processing, in-processing and post-processing algorithms. Pre-processing (Quadrianto et al., 2019; Ryu et al., 2018) methods focus on curating the dataset that includes removal of sensitive information or balancing the datasets. In-processing methods (Hashimoto et al., 2018b; Agarwal et al., 2018; Zafar et al., 2019; Lahoti et al., 2020; Prost et al., 2019; Veldanda et al., 2023a; Sohoni et al., 2020) alter the training mechanism by adding fairness constrains to the loss function or by training an adversarial framework to make predictions independent of sensitive attributes (Zhang et al., 2018). Post-processing methods (Hardt et al., 2016; Wang et al., 2020b; Savani et al., 2020) alter the outputs, for *e.g.* use different threshold for different sensitive attributes. In this work, we focus on *in-processing* algorithms.

Prior in-processing algorithms, including the ones referenced above, assume access to sensitive attributes on the training data and validation dataset. Recent work sought to train fair models without training data annotations (Liu et al., 2021; Nam et al., 2020; Hashimoto et al., 2018a; Creager et al., 2021; Levy et al., 2020; Duchi & Namkoong, 2018; Nam et al., 2022; Zhang et al., 2022; LaBonte et al., 2023) but require sensitive attributes on validation dataset to tune the hyperparameters. With Antigone, we seek to remove this restriction. Since JTT already compared against Nam et al. (2020) and Levy et al. (2020), a scalable version of (Duchi & Namkoong, 2018), we compare against JTT, although we believe Antigone can be used with these methods also.

Recently, Kirichenko et al. (2022) proposed a simple and light-weight framework to improve WGA. They demonstrated that re-training the last layer of an ERM model with a small, group-balanced dataset is sufficient for achieving robustness to spurious correlations. But, ground-truth SA's are required to balance the groups and tune-hyperparameters. AFR (Qiu et al., 2023) addresses this problem by retraining the last layer of an ERM model with a weighted loss that emphasizes the examples where the ERM model under-performs, automatically up-weighting the disadvantaged group without ground-truth SA labels. AFR showed that it can achieve the same target label accuracy and WGA as DFR, but AFR still requires ground-truth SA on validation dataset to tune-hyper-parameters. We show that using Antigone's PSA, we can get close to the performance of AFR without considering any access to ground-truth SA labels.

GEORGE (Sohoni et al., 2020) and and ARL (Lahoti et al., 2020) are two methods that like Antigone do not require PSAs on validation data. Qualitative and empirical comparisons in section 4 show that Antigone outperforms both. ARL is demonstrated to be effective only on smaller datasets and on simple structured network architectures. Prior works (Sohoni et al., 2020; Wang, 2022) have also noted ARL does not scale well to complex network architectures (e.g.: ResNet18) and large vision datasets. On the other hand, both methods can account for multiple subgroups, although as we noted before, Antigone still helps improve fairness for both. Another line of work makes implicit assumptions on ground-truth sensitive attributes. Zhu et al. (2023) is based on open-source proxy models. These proxy models are trained on distinct datasets to predict sensitive attributes. Subsequently, such off-the-shelf proxy models (Alao et al., 2021) are employed to predict unknown sensitive attributes within the dataset. Zhao et al. (2022) presume that the proxy features are known a-priori, thereby gaining knowledge of the attributes on which the model discriminates. However, as noted in section 1, we consider a more realistic setting where no such knowledge is given/known a-priori. There are also some *post-processing* methods to improve fairness without access to sensitive attributes but assuming a small set of labelled data for auditing Kim et al. (2019). One could use Antigone to create this auditing dataset, albeit with noise. Evaluating Antigone with these post-processing methods is an avenue for future work.s

Finally, a parallel body of work has looked at fairness with noisy sensitive attributes and incomplete information from a theoretical perspective using simplified but representative mathematical models (Lamy et al., 2019; Wang et al., 2020a; Awasthi et al., 2021; Celis et al., 2021), and sometimes with restrictions on the classifier types, etc. Yet these methods have largely not been translated to practical implementations on large datasets and state-of-art deep networks, which is Antigone's end goal. Antigone is one of the first

methods to bring insights from this line of work to bear on practical implementations. As a side note, we observe that Lemma 2.2 adds an extra result that might be of interest to this stream of research.

## 7 Conclusion

We propose Antigone, a method to enable hyper-parameter tuning for fair ML models without access to sensitive attributes on training or validation sets. Antigone generates high-quality PSA labels by training a family of ERM models and using correctly (incorrectly) classified examples as proxies for majority (minority) group membership. We propose a hyperparameter free approach to pick the ERM models that obtains the most accurate PSA labels, and provide theoretical justification for this choice using the ideal MC noise model. Antigone produces more accurate sensitive attributes estimates compared to the state-of-art, and can be used to effectively tune hyperparameters of state-of-art fairness methods. Future work will also seek to address the variance in fairness metrics (Mozannar et al., 2020) introduced by finite sample size under the ideal MC noise model, extend Antigone to non-binary sensitive attributes, and use methods such as Autotune (Koch et al., 2018) to automatically search over larger hyper-parameter spaces to maximize the EDM metric.

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

# Appendix

## Availability

Code with README.txt file is available at: `https://github.com/akshajkumarv/fairness_without_demographics`

## A  Experimental Setup

### A.1  Compute

We trained all the models employing the JTT approach using Quadro RTX8000 (48 GB) NVIDIA GPU cards, whereas, for both GEORGE and ARL approaches, we used GeForce RTX3090 (24 GB) NVIDIA GPU cards.

### A.2  CelebA Dataset

**Dataset details:** CelebA (Liu et al., 2015) is an image dataset, consisting of 202,599 celebrity face images annotated with 40 attributes including gender, hair colour, age, smiling, etc. The task is to predict hair color, which is either blond $Y = 1$ or non-blond $Y = 0$ and the sensitive attribute is gender $A = \{\text{Men}, \text{Women}\}$. The dataset is split into training, validation and test sets with 162770, 19867 and 19962 images, respectively. Only 15% of individuals in the dataset are blond, and only 6% of blond individuals are men. Consequently, the baseline ERM model under-performs on the blond men.

**Hyper-parameter settings:** In all our experiments using CelebA dataset, we fine-tune a pre-trained ResNet50 architecture for a total of 50 epochs using SGD optimizer and a batch size of 128. We tune JTT over the same hyper-parameters as in their paper: three pairs of learning rates and weight decays, $(1e-04, 1e-04), (1e-04, 1e-02), (1e-05, 1e-01)$ for both stages, and over ten early stopping points up to $T = 50$ and $\lambda \in \{20, 50, 100\}$ for stage 2. For Antigone, we explore over the same learning rate and weight decay values, as well as early stopping at any of the 50 training epochs. We report results for DP, EO and WGA fairness metrics. In each case, we seek to optimize fairness while constraining average target label accuracy to ranges $\{[90, 91), [91, 92), [92, 93), [93, 94), [94, 95)\}$.

### A.3  Waterbirds Dataset

**Dataset details:** Waterbirds is a synthetically generated dataset, containing 11,788 images of water and land birds overlaid on top of either water or land backgrounds (Sagawa* et al., 2020). The task is to predict the bird type, which is either a waterbird $Y = 1$ or a landbird $Y = 0$ and the sensitive attribute is the background $A = \{\text{Water background}, \text{Land background}\}$. The dataset is split into training, validation and test sets with 4795, 1199 and 5794 images, respectively. While the validation and test sets are balanced within each target class, the training set contains a majority of waterbirds (landbirds) in water (land) backgrounds and a minority of waterbirds (landbirds) on land (water) backgrounds. Thus, the baseline ERM model under-performs on the minority group.

**Hyper-parameter settings:** In all our experiments using Waterbirds dataset, we fine-trained ResNet50 architecture for a total of 300 epoch using the SGD optimizer and a batch size of 64. We tune JTT over the same hyper-parameters as in their paper: three pairs of learning rates and weight decays, $(1e-03, 1e-04), (1e-04, 1e-01), (1e-05, 1.0)$ for both stages, and over 14 early stopping points up to $T = 300$ and $\lambda \in \{20, 50, 100\}$ for stage 2. For Antigone, we explore over the same learning rate and weight decay values, as well as early stopping points at any of the 300 training epochs. In each case, we seek to optimize fairness while constraining average accuracy to ranges $\{[94, 94.5), [94.5, 95), [95, 95.5), [95.5, 96), [96, 96.5)\}$.

In case of GEORGE, for both CelebA and Waterbirds datasets, we use the same architecture and early stopping stage 2 hyper-parameters ($T = 50$ for CelebA and $T = 300$ for Waterbirds) reported in their paper. For Antigone+GEORGE, we replace GEORGE's stage 1 with the Antigone model, which is identified by

searching over the same hyper-parameter space as in Antigone+JTT. To establish statistical significance and determine if Antigone+GEORGE's performance is significantly greater than GEORGE by itself, we conducted a paired statistical t-test. The null hypothesis (H0) states that the mean of Antigone+GEORGE is less than or equal to the mean of GEORGE, while the alternative hypothesis (H1) states that the mean of Antigone+GEORGE is greater than that of GEORGE.

### A.4   UCI Adult Dataset

**Dataset details:** Adult dataset (Dua & Graff, 2017) is used to predict if an individual's annual income is $\leq 50K$ ($Y = 0$) or $> 50K$ ($Y = 1$) based on several continuous and categorical attributes like the individual's education level, age, gender, occupation, etc. The sensitive attribute is gender $A = \{\text{Men}, \text{Women}\}$ Zemel et al. (2013). The dataset consists of 45,000 instances and is split into training, validation and test sets with 21112, 9049 and 15060 instances, respectively. The dataset has twice as many men as women, and only 15% of high income individuals are women. Consequently, the baseline ERM model under-performs on the minority group.

**Hyper-parameter settings:** In all our experiments using Adult dataset, we train a multi-layer neural network with one hidden layer consisting of 64 neurons. We train for a total of 100 epochs using the SGD optimizer and a batch size of 256. We tune Antigone and JTT by performing grid search over learning rates $\in \{1e-03, 1e-04, 1e-05\}$ and weight decays $\in \{1e-01, 1e-03\}$. For JTT, we explore over $T \in \{1, 2, 5, 10, 15, 20, 30, 35, 40, 45, 50, 65, 80, 95\}$ and $\lambda \in \{5, 10, 20\}$. In each case, we seek to optimize fairness while constraining average accuracy to ranges $\{[82, 82.5), [81.5, 82), [81, 81.5), [80.5, 81), [80, 80.5)\}$.

For ARL, we train for a total of 100 epochs. We choose the best learning rate and batch size by exploring all possible hyper-parameters for the learner and adversary in the hyper-parameter search space given by batch size $\in \{32, 64, 128, 256, 512\}$ and learning rate $\in \{0.001, 0.01, 0.1, 1, 2, 5\}$, as in their paper. The learner is a fully connected two layer feed-forward network with 64 and 32 hidden units in the hidden layers, with ReLU activation function. The adversary is a fully connected one layer feed-forward network with 32 hidden units in the single hidden layer, with ReLU activation function. For Antigone+ARL, we use Antigone's pseudo sensitive attribute labels, for hyper-parameter tuning, identified by searching over the same hyper-parameter space as in Antigone+JTT. We seek to optimize fairness while constraining average accuracy to ranges $\{[84.5, 85), [84, 84.5), [83.5, 84), [83, 83.5), [82.5, 83), [82, 82.5)\}$. A paired statistical t-test is used to establish statistical significance and determine if the fairness of Antigone+ARL is significantly greater than that of ARL alone. The null hypothesis (H0) states that the mean fairness of Antigone+ARL is less than or equal to that of ARL, while the alternative hypothesis (H1) states that the mean fairness of Antigone+ARL is greater.

## B   Quality of Antigone's Sensitive Attribute Labels

Antigone seeks to generate accurate PSA labels on validation data based on the EDM criterion (Lemma 2.2). In Appendix Figure 3(b), we empirically validate Lemma 2.2 by plotting EDM and noise parameters $\alpha_1$ (contamination in advantaged group), $\beta_1$ (contamination in disadvantaged group) and $1 - \alpha_1 - \beta_1$ (proportionality constant between true and estimated fairness) on CelebA dataset. From the figure, we observe that the EDM metric indeed captures the trend in $1 - \alpha_1 - \beta_1$, enabling early stopping at an epoch that minimizes contamination. The best early stopping point that maximizes EDM also has a very high $1 - \alpha_1 - \beta_1$.

Lemma 2.2 shows that the EDM metric is, in theory, proportional to $1 - \alpha - \beta$. To further validate this lemma, we consider a larger hyper-parameter space, including learning rate, weight decay, and early stopping epochs, resulting in a total of 900 data points from our Waterbirds dataset. We compute the Pearson correlation coefficient between the EDM metric and $1 - \alpha - \beta$ across these 900 different data points. The Pearson correlation coefficients are 0.90 and 0.95 for the waterbirds and landbirds classes, respectively, indicating a strong positive correlation between the EDM metric and $1 - \alpha - \beta$. In Appendix Figure 4, we validate Lemma 2.2 by plotting the EDM metric vs. $1 - \alpha - \beta$ for different hyper-parameter settings.

In Appendix Table 6, we compare Antigone's PSA label's precision, recall, and F1 scores to GEORGE with the baseline $k = 5$ and with $k = 2$ clusters on CelebA and Waterbirds. We find that Antigone outperforms

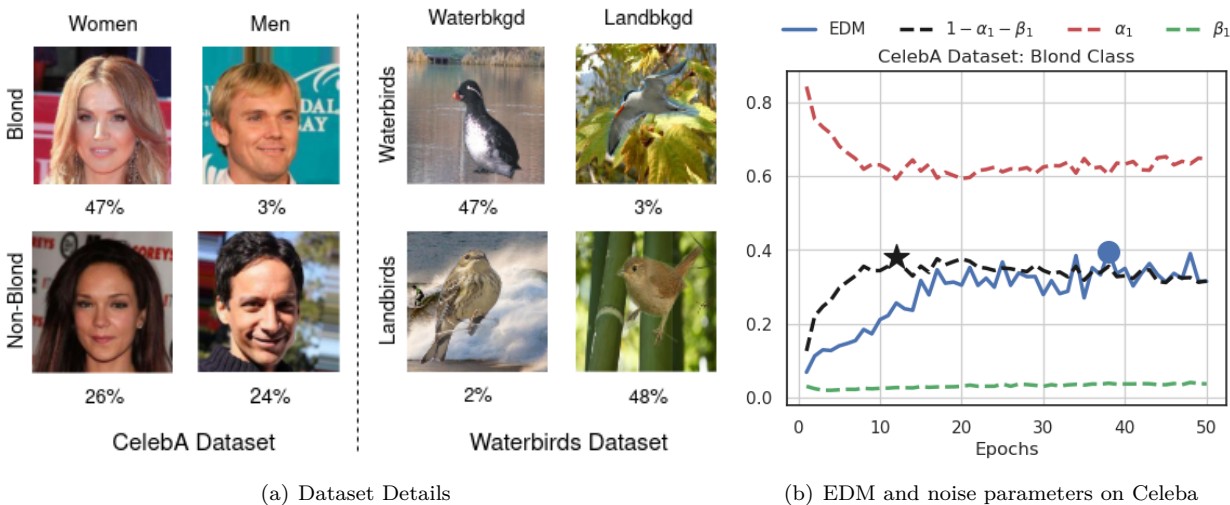

(a) Dataset Details

(b) EDM and noise parameters on Celeba

Figure 3: Figure (a) illustrates CelebA and Waterbirds datasets along with fraction of each sub-group examples in their respective training dataset. Figure (b) shows Euclidean Distance between Means (EDM) and noise parameters $\alpha_1, \beta_1$ and and $1 - \alpha_1 - \beta_1$ for the positive target class of CelebA dataset. The noise parameters are unknown in practice. Blue dot indicates the model that we pick to generate pseudo sensitive attributes, while black star indicates the model that maximizes $1 - \alpha_1 - \beta_1$.

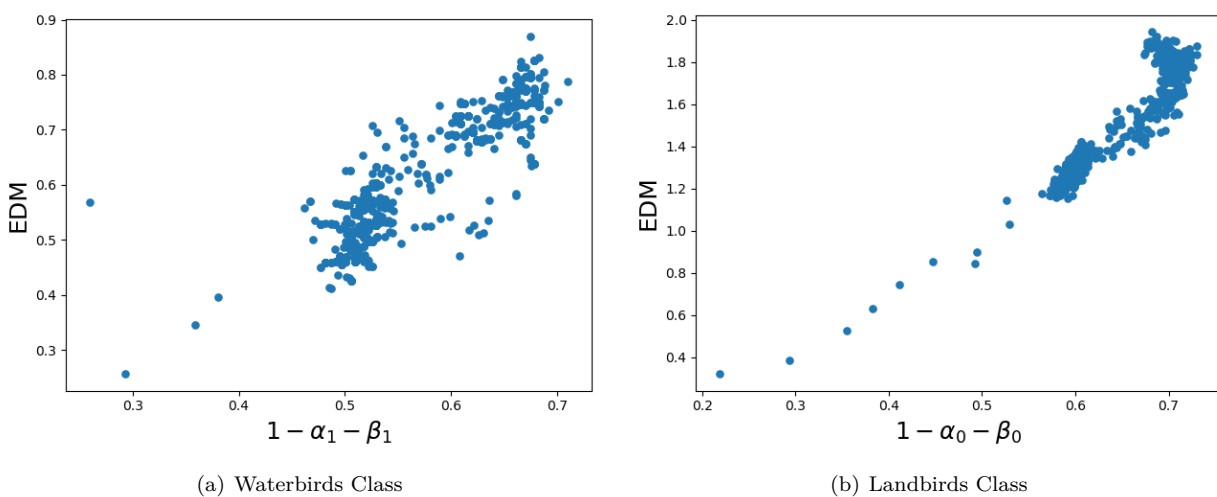

(a) Waterbirds Class

(b) Landbirds Class

Figure 4: Figure illustrates a strong positive correlation between EDM and $1 - \alpha - \beta$ with a Pearson correlation coefficient of 0.90 and 0.95 for the (a) waterbirds and (b) landbirds classes, respectively, on the Waterbirds dataset.

GEORGE. We also study the impact of varying the fraction of disadvantaged group individuals from 5%, 20%, 35%, 50% in the CelebA dataset (see Appendix Table 7). As the dataset gets more balanced, the models themselves are more fairer, and PSA label accuracy reduces (as expected). Nonetheless, disadvantaged group individuals are over-represented in incorrect sets for up to 35% imbalance.

Table 6: We tabulate the precision, recall, F1-score of the noisy validation groups generated from ERM model, GEORGE, GEORGE with number of clusters = 2 and Antigone. We observe that Antigone has higher precision and F1 scores across different noisy groups on CelebA and Waterbirds, respectively.

| | Antigone (w/o EDM) | GEORGE | GEORGE ($k = 2$) | Antigone (w/ EDM) |
|---|---|---|---|---|
| | CelebA (Precision, Recall, F1 Scores) | | | |
| Blond Men | 0.26, 0.31, 0.28 (0.02, 0.03, 0.01) | 0.09, 0.32, 0.13 (0.01, 0.09, 0.02) | 0.06, 0.70, 0.12 (0.01, 0.05, 0.01) | 0.36, 0.34, 0.35 (0.05, 0.04, 0.04) |
| Blond Women | 0.95, 0.94, 0.95 (0.01, 0.01, 0.01) | 0.94, 0.28, 0.43 (0.01, 0.04, 0.04) | 0.95, 0.35, 0.51 (0.00, 0.02, 0.02) | 0.96, 0.96, 0.96 (0.0, 0.0, 0.0) |
| Non-blond Women | 0.82, 0.13, 0.22 (0.01, 0.01, 0.02) | 0.51, 0.36, 0.42 (0.00, 0.01, 0.01) | 0.5, 0.76, 0.6 (0.00, 0.01, 0.01) | 0.86, 0.13, 0.22 (0.01, 0.01, 0.01) |
| Non-blond Men | 0.52, 0.97, 0.67 (0.00, 0.00, 0.01) | 0.53, 0.33, 0.40 (0.01, 0.02, 0.02) | 0.47, 0.23, 0.31 (0.01, 0.01, 0.01) | 0.52, 0.98, 0.68 (0.0, 0.0, 0.0) |
| CelebA Accuracy | $0.59 \pm 0.01$ | $0.33 \pm 0.01$ | $0.48 \pm 0.00$ | $0.60 \pm 0.00$ |
| | Waterbirds (Precision, Recall, F1 Scores) | | | |
| Waterbirds Landbkgd | 0.94, 0.26, 0.41 (0.01, 0.02, 0.02) | 0.56, 0.34, 0.43 (0.03, 0.02, 0.02) | 0.48, 0.57, 0.52 (0.01, 0.01, 0.01) | 0.96, 0.63, 0.76 (0.01, 0.04, 0.03) |
| Waterbirds Waterbkgd | 0.57, 0.98, 0.72 (0.01, 0.00, 0.00) | 0.55, 0.27, 0.36 (0.07, 0.03, 0.02) | 0.48, 0.39, 0.43 (0.02, 0.02, 0.02) | 0.73, 0.97, 0.83 (0.02, 0.01, 0.01) |
| Landbirds Waterbkgd | 0.96, 0.42, 0.58 (0.00, 0.03, 0.02) | 0.57, 0.36, 0.44 (0.04, 0.03, 0.03) | 0.55, 0.55, 0.55 (0.03, 0.03, 0.03) | 0.97, 0.65, 0.78 (0.00, 0.05, 0.04) |
| Landbirds Landbkgd | 0.63, 0.98, 0.76 (0.01, 0.00, 0.01) | 0.55, 0.24, 0.34 (0.04, 0.04, 0.02) | 0.55, 0.56, 0.55 (0.03, 0.04, 0.03) | 0.74, 0.98, 0.84 (0.03, 0.00, 0.02) |
| Waterbirds Accuracy | $0.68 \pm 0.01$ | $0.30 \pm 0.02$ | $0.53 \pm 0.03$ | $0.81 \pm 0.02$ |

Table 7: We tabulate the precision, recall, F1-score, *pseudo label accuracy* of the noisy validation groups generated by varying the fraction of minority group examples in each class of CelebA dataset. We observe that Antigone has higher precision, recall, F1 score and *pseudo label accuracy* if the imbalance is more in the training dataset.

| Fraction Minority | 5% | 20% | 35% | 50% |
|---|---|---|---|---|
| | Precision, Recall, F1 Score | | | |
| Blond Men (Minority) | 0.57, 0.40, 0.47 | 0.81, 0.17, 0.29 | 0.67, 0.15, 0.24 | 0.75, 0.14, 0.24 |
| Blond Women (Majority) | 0.97, 0.98, 0.98 | 0.83, 0.99, 0.90 | 0.68, 0.96, 0.79 | 0.53, 0.95, 0.68 |
| $1 - \alpha_1 - \beta_1$ | 0.54 | 0.64 | 0.35 | 0.28 |
| Blond Ps. Acc | 0.95 | 0.83 | 0.68 | 0.55 |
| Non-blond Women (Minority) | 0.45, 0.26, 0.33 | 0.59, 0.17, 0.26 | 0.63, 0.18, 0.28 | 0.63, 0.16, 0.26 |
| Non-blond Men (Majority) | 0.96, 0.98, 0.97 | 0.82, 0.97, 0.89 | 0.68, 0.94, 0.79 | 0.52, 0.91, 0.66 |
| $1 - \alpha_0 - \beta_0$ | 0.41 | 0.41 | 0.31 | 0.15 |
| Non-blond Ps. Acc. | 0.94 | 0.81 | 0.67 | 0.54 |
| Overall Ps. Acc. | 0.95 | 0.81 | 0.67 | 0.54 |

## B.1 Antigone's Performance on Large Pre-trained Models

Antigone's hyper-parameter search can include model architectures. We perform additional experiments by fine-tuning a pre-trained state-of-art large transformer model, ViT-B/16 Dosovitskiy et al. (2020), on the Waterbirds dataset and show that Antigone-ViT-B/16 can generate high quality PSA labels and improve down-stream fairness. We seek to maximize the EDM metric by exhaustively performing a grid search

over the hyper-parameter space which includes learning rate $\in \{1e-03, 1e-04, 1e-05\}$, weight decay $\in \{0, 1e-04, 1e-01, 1.0\}$, batch size $\in \{64, 512\}$, and gradient clipping $\in \{$False, True with threshold $= 1.0\}$.

In Appendix Table 8, we compare the quality of Antigone-ViT-B/16's PSA labels' F1 Score and PSA labels' accuracy to GEORGE with the baseline k = 5 and with k = 2 clusters. We find that Antigone-ViT-B/16 outperforms GEORGE. The table also reports results on a version of Antigone-ViT-B/16 that uses standard ERM training instead of EDM (Antigone-ViT-B/16 (w/o EDM)). We find that Antigone-ViT-B/16 (w/ EDM) provides higher pseudo-label accuracy compared to this baseline. To validate Lemma 2.2, we compute the Pearson correlation coefficient between the EDM metric and $1-\alpha-\beta$ over a larger hyper-parameter space containing 14,400 different data points from our Waterbirds datasets. The Pearson correlation coefficient is 0.88 and 0.97 for waterbirds and landbirds classes, respectively, indicating a strong positive correlation between the EDM metric and $1-\alpha-\beta$.

Table 8: F1 scores and *pseudo label accuracies* (Ps. Acc.) of Antigone-ViT-B/16 and GEORGE on Waterbirds dataset. We mark the best performance in bold. WL (waterbirds landbkgd), WW (waterbirds waterbkgd), LW (landbirds waterbkgd) and LL (landbirds landbkgd) for Waterbirds dataset.

|  | Antigone-ViT-B/16 (w/o EDM) | GEORGE | GEORGE ($k=2$) | Antigone-ViT-B/16 (w/ EDM) |
|---|---|---|---|---|
|  | Waterbirds (F1 Scores) | | | |
| WL | $0.28_{\pm 0.01}$ | $0.43_{\pm 0.02}$ | $0.52_{\pm 0.01}$ | $\mathbf{0.71}_{\pm 0.07}$ |
| WW | $0.69_{\pm 0.01}$ | $0.36_{\pm 0.02}$ | $0.43_{\pm 0.02}$ | $\mathbf{0.67}_{\pm 0.04}$ |
| LW | $0.18_{\pm 0.05}$ | $0.44_{\pm 0.03}$ | $0.55_{\pm 0.03}$ | $\mathbf{0.64}_{\pm 0.02}$ |
| LL | $0.68_{\pm 0.01}$ | $0.34_{\pm 0.02}$ | $0.55_{\pm 0.03}$ | $\mathbf{0.77}_{\pm 0.00}$ |
| Ps. Acc. | $0.57_{\pm 0.02}$ | $0.30_{\pm 0.02}$ | $0.53_{\pm 0.03}$ | $\mathbf{0.72}_{\pm 0.00}$ |

In Appendix Table 9, we compare the test target label accuracies and WGA achieved by Antigone-ViT-B/16 + AFR vs. Ground-Truth+AFR, where we utilize ResNet-50 as the backbone model for AFR. Furthermore, in Appendix Table 10, we also compare the test target label accuracies and WGA achieved by Antigone-ViT-B/16 + AFR-ViT-B/16 vs. Ground-Truth+AFR-ViT-B/16, where we utilize ViT-B/16 as the backbone model for AFR-ViT-B/16. The results are consistent with prior observations and show that Antigone-ViT-B/16 can be successfully used with large pre-trained models.

Table 9: Performance of AFR using Antigone-ViT-B/16's validation PSA compared with Ground-Truth + AFR on Waterbirds dataset. Higher WGA is better. We observe that Antigone-ViT-B/16 + AFR is close to the performance of Ground-Truth + AFR.

|  | Waterbirds | |
|---|---|---|
| Method | Avg Acc | WGA |
| ERM | $0.98_{\pm 0.02}$ | $0.64_{\pm 0.01}$ |
| Antigone-ViT-B/16 + AFR | $0.90_{\pm 0.05}$ | $0.83_{\pm 0.02}$ |
| Ground-Truth + AFR | $0.93_{\pm 0.03}$ | $0.84_{\pm 0.02}$ |

Table 10: Performance of AFR-ViT-B/16 using Antigone-ViT-B/16's validation PSA compared with Ground-Truth + AFR-ViT-B/16 on Waterbirds dataset. Higher WGA is better. We observe that Antigone-ViT-B/16 + AFR-ViT-B/16 is close to the performance of Ground-Truth + AFR-ViT-B/16.

| Method | Waterbirds | |
| --- | --- | --- |
| | Avg Acc | WGA |
| ERM | $0.99_{\pm 0.00}$ | $0.82_{\pm 0.01}$ |
| Antigone-ViT-B/16 + AFR-ViT-B/16 | $0.97_{\pm 0.01}$ | $0.93_{\pm 0.01}$ |
| Ground-Truth + AFR-ViT-B/16 | $0.97_{\pm 0.00}$ | $0.93_{\pm 0.01}$ |

## C   Antigone + JTT

Appendix Table 11 and Appendix Table 12 show the performance of Antigone in conjunction with JTT on Waterbirds and UCI Adult datasets, respectively. While we observed on the validation set that Ground-Truth+JTT consistently exhibits higher (or equal) fairness compared to Antigone+JTT, it is important to note that fairness measurements may differ on test sets due to potential distribution shifts Xu & Goodacre (2018).

To show that Antigone is effective on multiple binary sensitive attributes, we evaluate the trained Antigone+JTT models with age as the sensitive attribute on the CelebA dataset. The baseline ERM model achieves target label accuracy of 95.83% and worst-group accuracy (WGA) of 78.14%. Antigone improves WGA to 86.23% with a target label accuracy of 95.25%; and improves WGA further to 91.22% with target label accuracy of 93.53%. Antigone similarly improves demographic parity gap (2.21% to 1.81%) and equal opportunity gap (4.88% to 2.59%) with target label accuracy >95%.

Table 11: We report the (Test average target label accuracy, Test fairness metric) for different validation accuracy thresholds on Waterbirds dataset. We observe that Antigone + JTT (our noisy sensitive attributes) improves fairness over baseline ERM model and closes the gap with Ground-Truth + JTT (ground-truth sensitive attributes).

| Val. Thresh. | Method | DP Gap | EO Gap | Worst-group |
|---|---|---|---|---|
| [96, 96.5) | Antigone + JTT | (95.8, 3.9) ± (0.4, 0.4) | (96.2, 10.7) ± (0.3, 7.1) | (96.3, 83.0) ± (0.4, 1.3) |
| | Ground-Truth + JTT | (95.8, 3.9) ± (0.4, 0.4) | (96.0, 7.1) ± (0.3, 1.4) | (96.3, 83.0) ± (0.4, 1.3) |
| [95.5, 96) | Antigone + JTT | (95.4, 2.8) ± (0.1, 0.1) | (96.0, 7.5) ± (0.2, 1.5) | (96.3, 83.2) ± (0.3, 0.6) |
| | Ground-Truth + JTT | (95.4, 2.9) ± (0.4, 1.1) | (95.6, 6.0) ± (0.3, 2.1) | (96.1, 83.5) ± (0.5, 0.8) |
| [95, 95.5) | Antigone + JTT | (94.5, 1.5) ± (0.6, 0.6) | (94.7, 4.2) ± (0.9, 3.1) | (94.7, 85.9) ± (0.9, 1.4) |
| | Ground-Truth + JTT | (94.4, 1.7) ± (0.7, 0.7) | (94.3, 1.1) ± (0.5, 0.6) | (95.1, 86.8) ± (0.6, 1.1) |
| [94.5, 95) | Antigone + JTT | (94.2, 0.4) ± (0.4, 0.4) | (93.8, 2.0) ± (0.5, 1.4) | (94.2, 86.7) ± (0.8, 1.8) |
| | Ground-Truth + JTT | (93.6, 0.6) ± (0.5, 0.5) | (93.8, 2.0) ± (0.5, 1.4) | (94.1, 88.2) ± (0.6, 0.7) |
| [94.0, 94.5) | Antigone + JTT | (93.0, 1.5) ± (0.6, 0.3) | (93.6, 4.8) ± (1.2, 3.0) | (93.7, 87.9) ± (0.5, 1.4) |
| | Ground-Truth + JTT | (93.1, 1.5) ± (0.3, 0.4) | (93.2, 4.0) ± (1.0, 2.1) | (93.8, 88.1) ± (0.7, 1.1) |
| | ERM | (97.3, 21.3) ± (0.2, 1.1) | (97.3, 35.0) ± (0.2, 3.4) | (97.3, 59.1) ± (0.2, 3.8) |

Table 12: (Avg. target label accuracy, Fairness) on test data for different validation accuracy thresholds on the UCI Adult dataset. Lower DP and EO gaps are better. Higher WGA is better.

| Val. Thresh. | Method | DP Gap | EO Gap | Worst-group Acc. |
|---|---|---|---|---|
| >=82 and <82.5 | Antigone + JTT | (81.85, 11.76) ± (0.11, 3.53) | (81.46, 2.68) ± (0.24, 5.67) | (81.65, 54.58) ± (0.20, 0.87) |
| | Ground-Truth + JTT | (81.83, 11.92) ± (0.18, 3.62) | (81.49, 3.14) ± (0.17, 4.97) | (81.65, 54.58) ± (0.16, 0.76) |
| >=81.5 and <82 | Antigone + JTT | (81.74, 11.72) ± (0.15, 3.48) | (81.65, 6.39) ± (0.18, 1.14) | (81.56, 56.32) ± (0.4, 1.22) |
| | Ground-Truth + JTT | (81.57, 10.97) ± (0.24, 3.87) | (81.75, 6.1) ± (0.37, 1.85) | (81.52, 57.24) ± (0.34, 1.32) |
| >=81 and <81.5 | Antigone + JTT | (81.01, 9.11) ± (0.19, 3.67) | (81.14, 1.43) ± (0.24, 1.25) | (81.19, 54.46) ± (0.25, 1.64) |
| | Ground-Truth + JTT | (81.05, 8.92) ± (0.25, 3.34) | (81.11, 2.63) ± (0.22, 1.50) | (81.07, 55.04) ± (0.34, 1.07) |
| >=80.5 and <81 | Antigone + JTT | (80.71, 8.36) ± (0.31, 2.54) | (80.57, 2.28) ± (0.35, 1.31) | (80.63, 56.3) ± (0.29, 1.81) |
| | Ground-Truth + JTT | (80.41, 7.04) ± (0.4, 2.97) | (80.87, 5.93) ± (0.28, 1.52) | (80.53, 56.23) ± (0.23, 0.98) |
| >=80 and <80.5 | Antigone + JTT | (80.09, 7.54) ± (0.26, 2.18) | (80.19, 2.54) ± (0.32, 1.39) | (80.18, 57.67) ± (0.22, 1.73) |
| | Ground-Truth + JTT | (80.09, 5.63) ± (0.39, 2.59) | (80.23, 6.84) ± (0.15, 1.85) | (79.88, 57.50) ± (0.44, 1.75) |
| | ERM | (84.82, 53.83) ± (0.09, 0.20) | (84.82, 9.70) ± (0.09, 0.75) | (84.82, 53.14) ± (0.09, 0.58) |

## D   Comparison with GEORGE

Appendix Figure 5 and Appendix Figure 6 shows the performance of both Antigone+GEORGE and GEORGE experiments across multiple runs on CelebA and Waterbirds datasets, respectively.

Table 13: Antigone + JTT vs Ideal MC Model + JTT (Avg. target label accuracy, Fairness) comparison on test data for different validation accuracy thresholds on the CelebA dataset. Lower DP and EO gaps are better. Higher WGA is better.

|  |  | DP Gap | EO Gap | WGA |
|---|---|---|---|---|
| [94, 95) | Antigone + JTT | (94.9, 14.7) | (94.6, 33.7) | (94.5, 61.7) |
|  | Ideal MC + JTT | (94.9, 14.7) | (94.4, 34.1) | (94.4, 58.3) |
| [93, 94) | Antigone + JTT | (93.7, 12.2) | (93.9, 30.3) | (93.3, 60.0) |
|  | Ideal MC + JTT | (93.7, 12.2) | (93.5, 26.3) | (93.7, 65.0) |
| [92, 93) | Antigone + JTT | (93.1, 12.1) | (92.4, 22.9) | (92.9, 65.6) |
|  | Ideal MC + JTT | (93.1, 12.1) | (93.0, 22.7) | (93.2, 69.4) |
| [91, 92) | Antigone + JTT | (91.9, 9.3) | (91.1, 13.9) | (91.1, 66.7) |
|  | Ideal MC + JTT | (91.9, 9.3) | (92.2, 19.1) | (91.8, 73.9) |
| [90, 91) | Antigone + JTT | (91.1, 7.9) | (91.1, 13.9) | (91.1, 66.7) |
|  | Ideal MC + JTT | (90.9, 8) | (90.4, 18.9) | (91.4, 72.2) |

# E    Comparison with ARL

Appendix Table 14 shows the performance of Antigone+ARL on UCI Adult dataset.

Table 14: Comparison of (Avg. target label accuracy, Fairness) between ARL using Antigone's noisy validation data, ARL alone, and ground-truth validation data on the UCI Adult dataset. Lower DP and EO gaps indicate better fairness, while higher WGA is better. Antigone + ARL consistently outperforms ARL across various validation accuracy thresholds and fairness metrics (as shown in bold). The $p$-values are marked with $*$ accordingly: $*$ for $p < 0.1$, $**$ for $p < 0.05$, $***$ for $p < 0.01$, and $****$ for $p < 0.001$.

| Val. Thresh. | Method | DP Gap | EO Gap | Worst-group Acc. |
|---|---|---|---|---|
| [84.5, 85) | Antigone + ARL | $(\mathbf{84.51, 16.84})^{**} \pm (0.13, 1.08)$ | $(\mathbf{84.1, 5.64})^{***} \pm (0.08, 1.43)$ | $(\mathbf{84.14, 59.68})^{***} \pm (0.13, 1.27)$ |
|  | ARL | $(84.53, 19.01) \pm (0.13, 0.97)$ | $(84.53, 9.35) \pm (0.13, 0.88)$ | $(84.53, 55.08) \pm (0.13, 3.03)$ |
|  | Ground-Truth + ARL | $(84.5, 15.86) \pm (0.24, 1.02)$ | $(84.43, 4.97) \pm (0.17, 1.23)$ | $(84.08, 62.66) \pm (0.32, 0.71)$ |
| [84, 84.5) | Antigone + ARL | $(\mathbf{84.12, 15.55})^{***} \pm (0.21, 1.14)$ | $(\mathbf{83.7, 6.65})^{**} \pm (0.22, 1.61)$ | $(\mathbf{83.47, 60.9})^{**} \pm (0.22, 2.1)$ |
|  | ARL | $(84.02, 18.35) \pm (0.17, 1.04)$ | $(84.02, 8.04) \pm (0.17, 1.21)$ | $(84.02, 55.69) \pm (0.17, 2.93)$ |
|  | Ground-Truth + ARL | $(84.23, 15.26) \pm (0.12, 0.9)$ | $(83.81, 5.57) \pm (0.12, 1.58)$ | $(83.62, 64.74) \pm (0.14, 0.65)$ |
| [83.5, 84) | Antigone + ARL | $(\mathbf{83.29, 15.13})^{***} \pm (0.21, 1.17)$ | $(\mathbf{83.07, 5.36})^{**} \pm (0.22, 2.86)$ | $(\mathbf{82.93, 62.17})^{***} \pm (0.13, 1.63)$ |
|  | ARL | $(83.54, 19.43) \pm (0.12, 1.17)$ | $(83.54, 10.52) \pm (0.12, 2.16)$ | $(83.54, 53.72) \pm (0.12, 2.66)$ |
|  | Ground-Truth + ARL | $(83.53, 14.7) \pm (0.19, 0.87)$ | $(83.17, 4.34) \pm (0.17, 0.69)$ | $(83.31, 66.61) \pm (0.24, 1.6)$ |
| [83, 83.5) | Antigone + ARL | $(\mathbf{82.98, 15.05})^{***} \pm (0.18, 1.45)$ | $(\mathbf{82.69, 5.4})^{*} \pm (0.16, 3.15)$ | $(\mathbf{82.55, 65.85})^{***} \pm (0.31, 0.58)$ |
|  | ARL | $(83.2, 18.24) \pm (0.13, 2.45)$ | $(83.2, 8.69) \pm (0.13, 3.68)$ | $(83.2, 54.21) \pm (0.13, 4.99)$ |
|  | Ground-Truth + ARL | $(82.86, 14.84) \pm (0.13, 1.26)$ | $(83.04, 6.03) \pm (0.13, 1.56)$ | $(82.47, 66.75) \pm (0.16, 1.56)$ |
| [82.5, 83) | Antigone + ARL | $(\mathbf{82.69, 14.16})^{****} \pm (0.17, 1.42)$ | $(\mathbf{82.32, 8.23})^{**} \pm (0.14, 2.84)$ | $(\mathbf{82.12, 66.93})^{****} \pm (0.16, 1.46)$ |
|  | ARL | $(82.71, 20.27) \pm (0.37, 0.62)$ | $(82.71, 10.32) \pm (0.37, 2.17)$ | $(82.71, 54.4) \pm (0.37, 3.28)$ |
|  | Ground-Truth + ARL | $(82.63, 13.79) \pm (0.18, 1.52)$ | $(82.21, 4.31) \pm (0.28, 3.04)$ | $(81.91, 69.08) \pm (0.21, 1.52)$ |
| [82, 82.5) | Antigone + ARL | $(\mathbf{82.3, 13.63})^{****} \pm (0.18, 0.69)$ | $(\mathbf{81.88, 8.76})^{**} \pm (0.16, 1.46)$ | $(\mathbf{81.39, 66.61})^{***} \pm (0.29, 1.97)$ |
|  | ARL | $(82.16, 21.63) \pm (0.33, 1.69)$ | $(82.16, 13.55) \pm (0.33, 2.98)$ | $(82.16, 52.82) \pm (0.33, 5.58)$ |
|  | Ground-Truth + ARL | $(82.34, 13.21) \pm (0.25, 1.07)$ | $(81.97, 3.39) \pm (0.38, 2.71)$ | $(81.35, 69.98) \pm (0.23, 1.38)$ |
|  | ERM | $(84.69, 18.27) \pm (0.08, 0.5)$ | $(84.69, 9.39) \pm (0.08, 1.11)$ | $(84.69, 53.36) \pm (0.08, 1.97)$ |

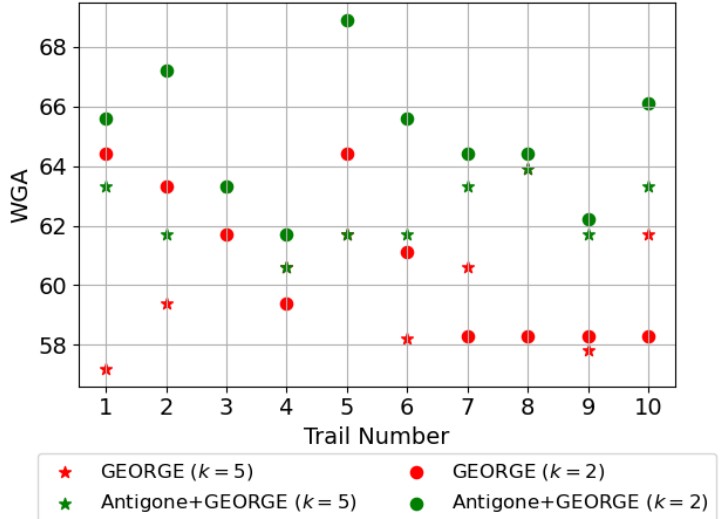

Figure 5: Figure illustrates the performance of Antigone+GEORGE and GEORGE in terms of target label accuracy and WGA across multiple trials for both $k = 5$ and $k = 2$ on the CelebA dataset.

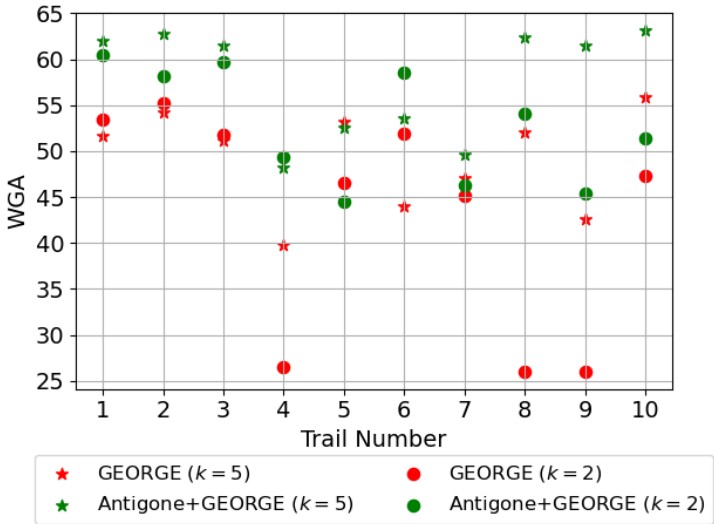

Figure 6: Figure illustrates the performance of Antigone+GEORGE and GEORGE in terms of target label accuracy and WGA across multiple trials for both $k = 5$ and $k = 2$ on the Waterbirds dataset.

