# OpenReview forum: "Hyper-parameter Tuning for Fair Classification without Sensitive Attribute Access"
_TMLR — Accepted by TMLR_

### Review · Reviewer_yigd · 2023-12-12

**Summary Of Contributions:**

In this paper, the authors propose an algorithmic framework to train fair classifiers, i.e., classifiers which correct bias arising from sensitive data attributes (SA) such as gender or race, under the assumption that these attributes are not provided at any stage of the procedure (either training or validation). In practice, this setting is motivated by the fact that data subjects may abstain from providing sensitive information when data is collected. Contrary to most of the previous works which assumed that the SA were given at validation phase to conduct hyper-parameter tuning [1,2], this approach provides a hyper-parameter tuning procedure which is more in line with reality. In particular, their method works with binary SA.

The authors propose the algorithm Antigone, which works in three steps. Given a range of hyper-parameter values, they first train an Empirical Risk Minimization (ERM) model for each of these values, which then provides proxies for the SA labels by comparing the prediction given by the ERM model to the true output (denoted PSA). Then, in validation phase, they select the model which maximizes the "Euclidean distance between the means" (EDM): this quantity simply refers to the squared distance between the mean of the pseudo-advantaged inputs (PSA-labeled 0) and the mean of the pseudo-disadvantaged inputs (PSA-labeled 1). This criterion consists in the main contribution of the paper and originates from the assumption of "mutually contamined" noise model, which states that the PSA labelization is a noisy version of the SA labelization [3]. Finally, the selected ERM model can be used as input to any method that trains fair models without access to SA on training data but requires SA for the validation. This algorithm works in the three commonly fairness settings: demographic parity, equal opportunity and worst-group accuracy.

In practice, the authors compare Antigone with state-of-the art methods that do not require  SA on the training dataset: JTT [1], which requires SA at validation phase; GEORGE [4] and ARL [5], which do not require SA at validation phase. They propose several numerical experiments in high-dimensional setting involving neural networks, where the hyper-parameter are learning rate, weight decay and early-stopping points. In particular, the results justify the use of their EDM criterion, as well as the superiority of their method.

[1] Just train twice: improving group robustness without training group information. Liu et al., 2021.

[2] Environment inference for invariant learning. Creager et al., 2021.

[3] Noise-tolerant fair classification. Lamy et al. 2019.

[4] No sub-class left behind [...]. Sohoni et al., 2020

[5] Fairness without demographics through adversarially reweighted learning. Lahoti et al., 2020.

**Audience:**

Yes

**Claims And Evidence:**

Yes

**Requested Changes:**

I would simply recommend the authors to conduct several experiments on the model of Figure 1, to validate Lemma 2.2 in multiple settings.

**Strengths And Weaknesses:**

**Strengths.** The main strength of this paper is the clarity of the approach. The method proposed here is simple, but well justified in theory with very good experimental results. Moreover, the authors put much effort in explaining the design of their experiments, the choice of their setup, and conducting fair comparison with previous methods.

**Weaknesses.** As noted by the authors, Antigone is limited by the use of binary sensitive attributes, which is a main issue for related methods. Moreover, the selection made at the validation phase requires to train as much models as the number of hyperparameter values: in the experiments, this is done for a small number of values. However, in practice, one could consider higher range, and in this case, the authors do not provide a strategy that could reduce the complexity of their approach.

---

> ### Author Response · Authors · 2024-01-16
> **Author Response to Reviewer yigd**
>
> **Q1) As noted by the authors, Antigone is limited by the use of binary sensitive attributes, which is a main issue for related methods.**
>
> R1)We agree this is a limitation currently. As noted  in Section 5, in its current form, Antigone only explicitly deals with binary sensitive attributes. In Section 5, we discuss some ideas on relaxing this assumption by further subdividing the incorrect set, or via multiple rounds of Antigone+JTT where in each round we address any remaining fairness gaps. We note also that although GEORGE addresses multiple (k) subgroups, tuning k is challenging and results for larger k are sometimes worse than k = 2.
>
> **Q2) The selection made at the validation phase requires training as many models as the number of hyperparameter values: in the experiments, this is done for a small number of values. However, in practice, one could consider a higher range, and in this case, the authors do not provide a strategy that could reduce the complexity of their approach.**
>
> R2)  The reviewer is correct that hyper-parameter search over large search spaces is computationally challenging. This is a challenge faced not only by Antigone, but also by all of the SOTA methods in this area even assuming access to ground-truth SAs on the validation set, or indeed as would be the case with any other machine learning task. Search over larger hyper-parameter spaces can be accomplished using methods such as Autotune [1] (and follow-up work) using the EDM metric as the reward function (we updated Section 7 to note this as part of our future work).
>
> [1] Koch, P., Golovidov, O., Gardner, S., Wujek, B., Griffin, J. and Xu, Y., 2018, July. Autotune: A derivative-free optimization framework for hyperparameter tuning. In Proceedings of the 24th ACM SIGKDD International Conference on Knowledge Discovery & Data Mining (pp. 443-452).
>
> **Q3) I would simply recommend the authors to conduct several experiments on the model of Figure 1, to validate Lemma 2.2 in multiple settings.**
>
> R3) Lemma 2.2 shows that the EDM metric is, in theory, proportional to $1-\alpha-\beta$. To validate Lemma 2.2, we compute the Pearson correlation coefficient between the EDM metric and $1-\alpha-\beta$ over a larger hyper-parameter space of learning rate, weight decay and early stopping epochs, containing a total of 900 different data points from our Waterbirds dataset. The Pearson correlation coefficient is 0.90 and 0.95 for waterbirds and landbirds classes, respectively, indicating a strong positive correlation between the EDM metric and $1-\alpha-\beta$. In Appendix Figure 4, we validate Lemma 2.2 by plotting the EDM metric vs. $1-\alpha-\beta$ for different hyper-parameter settings.
>
> We also observe similar results on a new experiment we ran on a pre-trained ViT-B/16 model on the Waterbirds dataset originally in response to Reviewer [j4jz](https://openreview.net/forum?id=ZSWKdRi2cU&noteId=ea5tssR9Ow). This involved exploring an even larger search space comprising 14,400 data points. The Pearson correlation coefficient for these 14,400 data points is 0.88 for the waterbirds class and 0.97 for the landbirds class.

---

### Review · Reviewer_46BP · 2023-12-21

**Summary Of Contributions:**

The paper proposes a new method (antigone) for training fair algorithms without access to sensitive attributes at either test or validation time. The core idea is to produce pseudo-sensitive attributes (PSA) by training many different ERM classification models with differing hyperparameters. We then define the PSA to be the labels associated with the ERM classifier that generates the largest difference in average feature values between the positive and negative class. These PSA can then be used as input to any model that estimates a fair classifier without access to sensitive attributes on the train set but requires access to sensitive attributes on the test set. The authors provide theoretical justification for antigone using the mutually contaminated noise model (Section 2.3). The authors report that the methods work well in experiments based on standard datasets.

**Audience:**

Yes

**Broader Impact Concerns:**

I have no concerns about the broader impact of this paper.

**Claims And Evidence:**

Yes

**Requested Changes:**

See my discussion of weakness above.

**Strengths And Weaknesses:**

Strengths:
1) Anti-gone is a simple procedure for constructing a fair classifier without access to sensitive attributes, who's main additional computational cost involves training potentially many ERM models with different hyper-parameters.

2) Anti-gone appears to work quite well in the provided experiments -- in particular, for example, JTT with antigone appears to perform nearly as well as JTT with ground-truth sensitive attribute labels.

Weaknesses:
1) The formal justification of the Antigone is based on an implausible mutually contaminated noise model (Section 2.3). It would be useful for the authors to provide some sense of the extent to which the mutually contaminated noise model may be violated in the empirical experiments. Concretely, the mutually contaminated noise model's key restriction is that P(A = a | X = x, PSA = p) does not vary across values of the feature (see Remark 2.3). Could you report in the experiments the extent to which this holds or does not hold? This is, in principle, a falsifiable assumption and it would be useful to see whether antigone's performance degrades as this is violated more.

2) Relatedly, I found the statement of the mutually contaminated noise model in Equation 7 difficult to interpret. Should that be a statement about the joint distribution of (X, A, PSA)? That is P(X | PSA = 1), P(X | PSA = 0) satisfies that decomposition?

2) The only illustration of different hyperparameters in the experiments appears to the be the number of epochs used for training and possible early start times. It would be useful/important to understand how antigone performs for other choices of function classes and/or hyperparameters. Among neural networks, there are many other hyperparameters we could consider such as the learning rate -- how does antigone perform when we only consider alternative learning rates? For example, many prediction models are simply based on threshold rules of a possibly regularized risk assessment -- can we apply antigone for the choice of regularization parameter? Random forests are often used for classification in fair machine learning, yet there are also many hyperparameters in the learning algorithm -- can we apply antigone for the choice of these hyperparameters? If antigone is tailored for the sorts of classifiers illustrated in the experiments, that should be clarified.

---

> ### Author Response · Authors · 2024-01-16
> **Author Response to Reviewer 46BP (Part 1/2)**
>
> **Q1) The formal justification of the Antigone is based on an implausible mutually contaminated noise model (Section 2.3). It would be useful for the authors to provide some sense of the extent to which the mutually contaminated noise model may be violated in the empirical experiments. Concretely, the mutually contaminated noise model's key restriction is that P(A = a | X = x, PSA = p) does not vary across values of the feature (see Remark 2.3). Could you report in the experiments the extent to which this holds or does not hold? This is, in principle, a falsifiable assumption and it would be useful to see whether antigone's performance degrades as this is violated more.**
>
> R1) This is an excellent point raised by the reviewer, and one that we had already addressed in the paper Appendix Table 12 and Ablation Studies (Section 4), but perhaps not highlighted clearly. Specifically, in Appendix Table 12, we compare Antigone+JTT against a *synthetically labeled validation dataset that exactly follows the ideal MC noise model.* We create this ideal MC noise model by extracting the ground-truth noise parameters from Antigone's PSA labels on the validation set, which we can do easily by assuming that ground-truth SA labels are known. We then use these noise parameters to create a new dataset consistent with the ideal MC noise, using the data generation process in Equation 7. We find that on DP Gap and EO Gap fairness metrics, Antigone’s results are comparable with those derived from the ideal MC model. On WGA, the most challenging fairness metric to optimize for, we find that the ideal MC model has a best-case WGA of 73.9% compared to Antigone’s 66.7% (both are substantially better than the ERM model’s WGA of 38%). This reflects the loss in fairness due to the gap between the assumptions of the idealized model versus Antigone’s implementation.
>
> We anticipate, as the reviewer noted, that the decline in fairness of Antigone will further grow as we move away from the ideal MC noise model. However, we show that Antigone works well on real-world datasets, even if they don’t follow the ideal MC noise model exactly. We note that we are not claiming the theoretical fairness guarantees of Lamy et al. (2019). Antigone is a practical solution to improve fairness like the other works we evaluate against; we use the MC model to guide our choice of the EDM metric and provide intuition as to its working.
>
> **Q2) Relatedly, I found the statement of the mutually contaminated noise model in Equation 7 difficult to interpret. Should that be a statement about the joint distribution of (X, A, PSA)? That is P(X | PSA = 1), P(X | PSA = 0) satisfies that decomposition?**
>
> R2) In Equation 7, we tried to follow the description in the original MC noise model paper [1] which deals with the data generation process. We agree with the reviewer that the following decomposition satisfies under the MC noise model:
> $$P(X | PSA = 1) = (1 − α) P(X | A=1) + α P(X | A=0)$$   $$P(X | PSA = 0) = β P(X | A=1) + (1 − β) P(X | A=0)$$
> and have noted this implication in the main draft of the paper. We thank the reviewer for the opportunity to clarify.
>
> [1] Scott, C., Blanchard, G. and Handy, G., 2013, June. Classification with asymmetric label noise: Consistency and maximal denoising. In Conference on learning theory (pp. 489-511). PMLR.

---

> ### Author Response · Authors · 2024-01-16
> **Author Response to Reviewer 46BP (Part 2/2)**
>
> **Q3) The only illustration of different hyperparameters in the experiments appears to the be the number of epochs used for training and possible early start times. It would be useful/important to understand how antigone performs for other choices of function classes and/or hyperparameters. Among neural networks, there are many other hyperparameters we could consider such as the learning rate -- how does antigone perform when we only consider alternative learning rates? For example, many prediction models are simply based on threshold rules of a possibly regularized risk assessment -- can we apply antigone for the choice of regularization parameter? Random forests are often used for classification in fair machine learning, yet there are also many hyperparameters in the learning algorithm -- can we apply antigone for the choice of these hyperparameters? If antigone is tailored for the sorts of classifiers illustrated in the experiments, that should be clarified.**
>
> R3) Indeed, Antigone can be applied to a large range of hyper-parameter choices, and all our existing experiments in paper tune over not just the early stopping epoch (as illustrated in Fig. 2), but also the learning rate and weight decay (see Section 3.3 for details of our hyper-parameter space). This includes our experiments on the UCI dataset where we perform a grid search over learning rates = {1e − 03, 1e − 04, 1e − 05}, weight decays = {1e − 01, 1e − 03}, and early stopping epochs (as described in Appendix A.4). We apologize if this was not already clear in the paper.
>
> Lemma 2.2 shows that the EDM metric is, in theory, proportional to $1-\alpha-\beta$. To further validate this lemma, we consider a larger hyper-parameter space, including learning rate, weight decay, and early stopping epochs, resulting in a total of 900 data points from our Waterbirds dataset. We compute the Pearson correlation coefficient between the EDM metric and $1-\alpha-\beta$ across these 900 different data points. The Pearson correlation coefficients are 0.90 and 0.95 for the waterbirds and landbirds classes, respectively, indicating a strong positive correlation between the EDM metric and $1-\alpha-\beta$. In Appendix Figure 4, we validate Lemma 2.2 by plotting the EDM metric vs. $1-\alpha-\beta$ for different hyper-parameter settings.
>
> Additionally, we performed new experiments (partly in response to Reviewer [j4jz](https://openreview.net/forum?id=ZSWKdRi2cU&noteId=ea5tssR9Ow)) by fine-tuning a pre-trained ViT-B/16 on the Waterbirds dataset on an even larger range of hyper-parameter settings and show that Antigone can generate high quality PSA labels and improve down-stream fairness. Specifically, we seek to maximize the EDM metric by exhaustively performing a grid search over the hyper-parameter search space which includes learning rate = {1e-03, 1e-04, 1e-05}, weight decay = {0, 1e-04, 1e-01, 1.0}, batch_size = {64, 512}, and gradient clipping = {False, True with threshold = 1.0}. In the following table, we compare the quality of Antigone’s PSA labels’ F1 Score and PSA labels’ accuracy to GEORGE with the baseline k = 5 and with k = 2 clusters. We find that Antigone outperforms GEORGE. To validate Lemma 2.2, we compute the Pearson correlation coefficient between the EDM metric and $1-\alpha-\beta$ over a larger hyper-parameter space containing 14,400 different data points from our Waterbirds datasets. The Pearson correlation coefficient is 0.88 and 0.97 for waterbirds and landbirds classes, respectively, indicating a strong positive correlation between the EDM metric and $1-\alpha-\beta$.
>
> |                        | GEORGE | GEORGE ($k = 2$) | Antigone (w/ EDM) |
> |------------------------|--------|-------------------|-------------------|
> | Waterbirds (F1 Scores)  |        |                   |                   |
> | WL                     | 0.43±0.02 | 0.52±0.01 | **0.71±0.07** |
> | WW                     | 0.36±0.02 | 0.43±0.02 | **0.67±0.04** |
> | LW                     | 0.44±0.03 | 0.55±0.03 | **0.64±0.02** |
> | LL                     | 0.34±0.02 | 0.55±0.03 | **0.77±0.00** |
> | Ps. Acc.               | 0.30±0.02 | 0.53±0.03 | **0.72±0.00** |
>
> In the table below, we compare the test target label accuracies and WGA achieved by Antigone+AFR vs. Ground-Truth+AFR. The results are consistent with prior observations and show that Antigone can be successfully used with large pre-trained models.
>
> | Method               | Waterbirds Avg Acc      | Waterbirds WGA          |
> |----------------------|-------------------------|-------------------------|
> | ERM                  | 0.98 ± 0.02             | 0.64 ± 0.01             |
> | Antigone + AFR  | 0.90 ± 0.05             | 0.83 ± 0.02             |
> | Ground-Truth + AFR   | 0.93 ± 0.03             | 0.84 ± 0.02             |

---

### Review · Reviewer_j4jz · 2023-12-30

**Summary Of Contributions:**

This paper studies the problem of fair machine learning by proposing a proxy-based method that first generates pseudo sensitive attributes on the validation data as proxy data. Here, the generation process can help the method not be restricted by the existing methods' assumptions that the sensitive attributes are known on the validation data. The authors conduct experiments on three datasets, and the results show superior performance of the proposed Autigone method.

**Audience:**

Yes

**Claims And Evidence:**

Yes

**Requested Changes:**

Please answer my questions, with corresponding changes, in the above-mentioned weaknesses.

**Strengths And Weaknesses:**

The strengths of this paper are as follows.

1. The studied problem is of great importance. The fairness problem is always a concern, especially for data involving sensitive attributes.
2. The proposed method achieves the best performance on three datasets.
3. The paper is written well.


The weaknesses of this paper are as follows.
1. The baselines are a bit out-of-date. There is only one baseline published after 2020 (JTT), and there is no recent method to be considered in the experiments. Actually, JTT has a lot of follow-up papers, and the authors should consider them both in the related work section and experiments section.
2. It is concerned whether the method can be generally and widely used. The proposed method should be evaluated on more backbone models, especially for these pretrained large models, for more extensive evaluation.
3. The predicted pseudo may still reveal the fairness issue.

---

> ### Author Response · Authors · 2024-01-16
> **Author Response to Reviewer j4jz (Part 1/2)**
>
> **Q1) The baselines are a bit out-of-date. There is only one baseline published after 2020 (JTT), and there is no recent method to be considered in the experiments. Actually, JTT has a lot of follow-up papers, and the authors should consider them both in the related work section and experiments section.**
>
> R1)  To address the reviewer’s concern, we performed additional experiments on the recently proposed Automatic Feature Reweighting (AFR) framework from ICML’23 [1] using the CelebA and Waterbirds dataset and updated the paper (changes highlighted in red) with results and discussion. *We observe that the results are consistent with prior observations and show that Antigone can be successfully applied on top of AFR, which has achieved SOTA results improving on methods like JTT and others.*
>
> AFR operates in two stages, training the model with 80% of the training data with an ERM objective, and using the remaining 20% of the data to fine-tune the last layer with a weighted fairness objective. The hyper-parameters of the second stage are tuned using a validation dataset with ground-truth SA labels. We refer to this as the Ground-Truth + AFR baseline. We substitute the ground-truth SA in the validation dataset with PSA acquired from Antigone (Antigone + AFR) and use it to tune AFR’s hyperparameters. A detailed description of AFR and the experimental setup can be found in Section 3 of the revised paper.
>
> In the table below (Section 4: Table 5 in the paper), we compare the test target label accuracies and WGA achieved by Antigone+AFR vs. Ground-Truth+AFR. Antigone+AFR is very close to Ground-Truth+AFR in terms of both target label accuracy and WGA, and substantially improves on standard ERM. On CelebA, Antigone+AFR improves WGA from 41% for standard ERM to 81% at the expense of a 5% target label accuracy drop. Ground-Truth + AFR improves WGA further up to 82% with a 4% target label accuracy drop. Similar observations hold for the Waterbirds dataset also.
>
> | Method            | CelebA Avg Acc | CelebA WGA | Waterbirds Avg Acc | Waterbirds WGA |
> |-------------------|----------------|------------|--------------------|----------------|
> | ERM               | 0.96 ± 0.01    | 0.41 ± 0.01 | 0.98 ± 0.02        | 0.64 ± 0.01    |
> | Antigone + AFR        | 0.91 ± 0.01    | 0.81 ± 0.01 | 0.92 ± 0.04        | 0.82 ± 0.02    |
> | Ground-Truth + AFR| 0.92 ± 0.00    | 0.82 ± 0.01 | 0.93 ± 0.03        | 0.84 ± 0.02    |
>
> Additionally, we have also updated the related works section (changes highlighted in red) to reflect several latest advancements in the field, including AFR.
>
> [1] Qiu, S., Potapczynski, A., Izmailov, P. and Wilson, A.G., 2023. Simple and Fast Group Robustness by Automatic Feature Reweighting. arXiv preprint arXiv:2306.11074.

---

> ### Author Response · Authors · 2024-01-16
> **Author Response to Reviewer j4jz (Part 2/2)**
>
> **Q2) It is concerned whether the method can be generally and widely used. The proposed method should be evaluated on more backbone models, especially for these pretrained large models, for more extensive evaluation.**
>
> R2) Antigone’s hyper-parameter search can include different model architectures. We perform additional experiments by fine-tuning a pre-trained state-of-art large transformer model, ViT-B/16, on the Waterbirds dataset and show that Antigone can generate high quality PSA labels and improve down-stream fairness. We updated the paper with this discussion in Appendix B.1 (changes highlighted in red).
>
> We seek to maximize the EDM metric by exhaustively performing a grid search over the hyper-parameter search space which includes learning rate = {1e-03, 1e-04, 1e-05}, weight decay = {0, 1e-04, 1e-01, 1.0}, batch_size = {64, 512}, and gradient clipping = {False, True with threshold = 1.0}. In the following table (Appendix Section B.1: Table 8 in the paper), we compare the quality of Antigone’s PSA labels’ F1 Score and PSA labels’ accuracy to GEORGE with the baseline k = 5 and with k = 2 clusters. We find that Antigone outperforms GEORGE.
>
> Table: F1 scores and pseudo label accuracies (Ps. Acc.) of Antigone and GEORGE on Waterbirds dataset. We mark the best performance in bold. WL (waterbirds landbkgd), WW (waterbirds waterbkgd), LW (landbirds waterbkgd) and LL (landbirds landbkgd) for Waterbirds dataset
> |                        | GEORGE | GEORGE ($k = 2$) | Antigone (w/ EDM) |
> |------------------------|--------|-------------------|-------------------|
> | Waterbirds (F1 Scores)  |        |                   |                   |
> | WL                     | 0.43±0.02 | 0.52±0.01 | **0.71±0.07** |
> | WW                     | 0.36±0.02 | 0.43±0.02 | **0.67±0.04** |
> | LW                     | 0.44±0.03 | 0.55±0.03 | **0.64±0.02** |
> | LL                     | 0.34±0.02 | 0.55±0.03 | **0.77±0.00** |
> | Ps. Acc.               | 0.30±0.02 | 0.53±0.03 | **0.72±0.00** |
>
> In the table below (Appendix Section B.1: Table 9 in the paper), we compare the test target label accuracies and WGA achieved by Antigone+AFR vs. Ground-Truth+AFR. The results are consistent with prior observations and show that Antigone can be successfully used with large pre-trained models.
>
> | Method               | Waterbirds Avg Acc      | Waterbirds WGA          |
> |----------------------|-------------------------|-------------------------|
> | ERM                  | 0.98 ± 0.02             | 0.64 ± 0.01             |
> | Antigone + AFR  | 0.90 ± 0.05             | 0.83 ± 0.02             |
> | Ground-Truth + AFR   | 0.93 ± 0.03             | 0.84 ± 0.02             |
>
> **Q3) The predicted pseudo may still reveal the fairness issue.**
>
> R3) We apologize, but we were not able to understand the gist of the reviewer’s request. We kindly request the reviewer to provide further clarification in their question which would better help us understand their concern and address it. Thank you.

---

### Author Response · Authors · 2024-01-16
**Rebuttal Submitted**

Dear Reviewers and Editors,

Thank you once again for offering valuable feedback on our manuscript. We have carefully addressed each of your concerns and made the necessary updates to the document. If you have any further questions, please don't hesitate to reach out. We are more than happy to assist.

Best Regards,

Authors

---

### Decision · Action_Editor_wdX3 · 2024-02-08

**Recommendation:** Accept with minor revision

**Comment:**

This paper introduces Antigone, a novel approach for training fair models without the need to access sensitive attributes during the validation phase. Instead of accessing these sensitive labels, Antigone creates pseudo reference labels from multiple different ERM classification models with varying hyperparameters. Reviewers agree that Antigone's methodology is both straightforward and clearly articulated (yigd, 46BP). Additionally, experiments indicate the method's applicability across diverse models (46BP, j4jz). During the rebuttal process, the authors provided further clarification on the interpretations of the equations and the experimental setup regarding the categories of hyperparameters considered. While the review by j4jz highlights concerns regarding the performance of the proposed methods when applied to recent backbone models, upon examining the paper, review comments, and rebuttals, the action editor read the paper&rebuttal and believes that some of these concerns are addressed in the rebuttal.  For instance, the authors have conducted additional analyses on AFR from ICML’23, as well as on various model architectures, including ViT-B/16, within their rebuttals. The authors are advised to expand upon this by including a comprehensive discussion and comparison with the backbone models, such as those developed in the follow-up works of JTT, in their camera-ready submission.

**Audience:**

Yes.

**Claims And Evidence:**

Yes.

---

> ### Author Response · Authors · 2024-03-06
> **Thank you for the reviews!**
>
> We would like to thank the reviewers and the Action Editor for their thoughtful insights. In addition to including the reviewer feedback, we would like to bring to your attention that we have made further modifications to the paper to include the results/discussion of AFR and performed additional experiments to demonstrate the effectiveness of Antigone on large pre-trained backbones. Specifically, we fine-tuned a pre-trained state-of-art large transformer model, ViT-B/16 (Dosovitskiy et al., 2020), on the Waterbirds dataset, resulting in target label accuracy of 99% and WGA of 82% for the ERM model (refer to Appendix B.1 for more details). In Appendix Table 8, we compare Antigone-ViT-B/16’s PSA labels to GEORGE with k = 5 and k = 2 clusters. We find that Antigone-ViT-B/16 outperforms GEORGE. Furthermore, we compare the test target label accuracies and WGA achieved by Antigone-ViT-B/16 + AFR vs. Ground-Truth+AFR (Appendix Table 10). The results are consistent with prior observations and show that Antigone-ViT-B/16 can be successfully used with large pre-trained models.